# CONVERGE FASTER, TALK LESS: HESSIAN-INFORMED FEDERATED ZEROTH-ORDER OPTIMIZATION

**Zhe Li**[1], **Bicheng Ying**[2], **Zidong Liu**[3], **Chaosheng Dong**[4], **Haibo Yang**[1]

[1]Rochester Institute of Technology, Rochester, NY 14623, USA

[2]Google, Los Angeles, CA 90034, USA

[3]ComboCurve Inc., Houston, TX 77005, USA

[4]Amazon.com, Seattle, WA 98109, USA

zl4063@rit.edu, ybc@google.com, z.liu@combocurve.com,
chaosd@amazon.com, hbycis@rit.edu

## ABSTRACT

Zeroth-order (ZO) optimization enables dimension-free communication in federated learning (FL), making it attractive for fine-tuning of large language models (LLMs) due to significant communication savings. However, existing ZO-FL methods largely overlook curvature information, despite its well-established benefits for convergence acceleration. To address this, we propose `HiSo`, a Hessian-informed ZO federated optimization method that accelerates convergence by leveraging global diagonal Hessian approximations, while strictly preserving scalar-only communication *without transmitting any second-order information*. Theoretically, for non-convex functions, we show that `HiSo` can achieve an accelerated convergence rate that is independent of the Lipschitz constant $L$ and model dimension $d$ under some Hessian approximation assumptions, offering a plausible explanation for the observed phenomenon of ZO convergence being much faster than its worst-case $\mathcal{O}(d)$-bound. Empirically, across diverse LLM fine-tuning benchmarks, HiSo delivers a 1~5× speedup in communication rounds over existing state-of-the-art ZO-FL baselines. This superior convergence not only cuts communication costs but also provides strong empirical evidence that Hessian information acts as an effective accelerator in federated ZO optimization settings.

## 1 INTRODUCTION

The explosive development of large language models (LLMs) has sparked strong interest in making their fine-tuning scalable across distributed and privacy-sensitive data sources (Naveed et al., 2023; Zhao et al., 2023). As a promising and effective paradigm, federated fine-tuning has been successfully applied to enable collaborative model training while preserving data privacy in a large amount of works (Kairouz et al., 2021; Cho et al., 2024; Li et al., 2025a; Zhang et al., 2025). Yet, modern LLMs' massive parameter scale (e.g., several billions) introduces a severe scalability barrier: due to communicating high-dimensional model updates, communication cost has become a primary bottleneck for federated LLM fine-tuning (Wu et al., 2025; Jia et al., 2025). For instance, fine-tuning a OPT-1.3B model by FedAvg (McMahan et al., 2017) usually requires about $1 \sim 5$ TB communication cost for one client (Li et al., 2025b). To overcome this burden, recent work has proposed using zeroth-order (ZO) optimization (Nesterov & Spokoiny, 2017) to enable dimension-free communication in FL. In particular, DeComFL (Li et al., 2025b) encodes both uplink and downlink communication using shared random seeds and scalar-only updates, achieving communication cost independent of model dimension. Specifically, it reduces the total communication cost from TB level to MB level. This dimension-free communication framework is especially attractive for federated LLM fine-tuning, where communication is a dominant bottleneck.

However, the practical effectiveness of ZO-based FL is limited by its seriously slow convergence (Fang et al., 2022; Li et al., 2025b). A key factor is that LLMs often exhibit heterogeneous and anisotropic curvature across their parameter space (Kingma & Ba, 2015; Yao et al., 2021; Benzing, 2022), making it difficult for vanilla ZO-SGD to adaptively scale updates. While prior work has shown that second-order information (e.g., Hessians or their diagonal approximations) can significantly

accelerate convergence (Kingma & Ba, 2015; Ye et al., 2018; Jiang et al., 2024; Zhao et al., 2025), estimating Hessian approximation and applying such curvature-aware techniques in FL are non-trivial and even more pronounced in dimension-free communication frameworks, where **transmitting any Hessian-related information reintroduces expensive costs that scale with model size linearly or even quadratically, directly contradicting the goal of scalar-only communication**. This tension leads to our key research question:

> *Q: Can we accelerate federated zeroth-order fine-tuning while preserving dimension-free communication?*

To answer this question, we first propose a generalized scalar-only communication FL framework that decouples scalar-only communication from its tight connection with vanilla ZO-SGD, enabling the integration of Hessian-informed optimization[1]. Within this framework, we are equipped to introduce `HiSo`, an efficient FL algorithm via Hessian-informed ZO optimization and Scalar-only communication. Specifically, it captures curvature information through diagonal Hessian approximation without increasing Hessian-related communication cost. `HiSo` maintains the scalar-only communication and significantly improves convergence via Hessian-informed preconditioning. Our theoretical and empirical results and contributions can primarily be summarized as follows:

- We propose a flexible FL framework with scalar-only communication in both uplink and downlink, which supports a broader class of optimization algorithms beyond vanilla ZO-SGD.

- Under this framework, we propose `HiSo`, a fast federated ZO optimization method via Hessian-informed zeroth-order optimization and Scalar-only communication. It utilizes global Hessian information to speed up convergence while preserving scalar-only communication (without the need to communicate Hessian-related information).

- Theoretically, we propose a novel condition to get a tight estimation of the variance of Hessian-informed ZO gradient under the low-effective rank and whitening assumptions. With this treatment, we prove that `HiSo` can achieve a convergence rate independent of model dimension and function smoothness in non-convex settings, marking the first such result for ZO methods in FL. In addition, our analysis generalizes the state-of-the-art DeComFL framework and, importantly, extends the theoretical guarantees to multiple local updates - a key component of practical federated learning that DeComFL does not support in its convergence analysis.

- Empirically, `HiSo` achieves up to $5\times$ faster convergence than DeComFL, while delivering higher test accuracy than all ZO baselines across all tasks. Compared to first-order baselines, up to 90 million times communication savings can be gained.

## 2 RELATED WORK

**Adaptive Gradient Methods & Hessian-Informed Zeroth-Order (ZO) Optimization.** To accelerate first-order FL, adaptive FL algorithms (e.g., FedAdam, FedYogi, FedAdagrad (Reddi et al., 2021)) have been introduced to address the slow convergence in heterogeneous environments. By adaptively adjusting learning rates or applying momentum techniques, these methods significantly outperform vanilla FedAvg in terms of convergence speed and final accuracy. Parallel to this line, recent ZO advances have shown its effectiveness in gradient-free learning, especially when gradients are unavailable or expensive to compute. To further enhance convergence speed and stability, several studies (Ye et al., 2018; Zhang et al., 2022; Chen et al., 2024; Zhao et al., 2025; Kim et al., 2025) proposed Hessian-informed ZO methods that incorporate second-order information, such as diagonal Hessian approximations, as preconditioning to improve the quality of gradient estimation and reduce variance, which shows the acceleration in single-node settings.

**Communication-Efficient Federated Learning & Scalar-Only Communication.** Communication efficiency is a critical challenge in FL primarily due to the frequent transmission of high-dimensional model updates between clients and the server (Kairouz et al., 2021; Jia et al., 2025). Numerous methods have been proposed to reduce communication overhead in FL, including parameter-efficient methods, such as Low-Rank Adaptation (LoRA) (Sun et al., 2024; Guo et al., 2025) to transmit

---

[1]The term "Hessian-informed" does not imply that we calculate the full Hessian matrix; rather, the update approximates the Hessian preconditioning direction, as we will explain in detail later.

only a low-rank trainable matrix representing model updates, and compression techniques used to reduce the size of transmitted data (Yang et al., 2021; Wang et al., 2022; Hönig et al., 2022; Su et al., 2024; Li et al., 2024; Zakerinia et al., 2024). Moreover, ZO optimization has also been introduced to the FL context (Fang et al., 2022; Qin et al., 2024; Liang et al., 2025). FedZO (Fang et al., 2022) integrates ZO-SGD into FL, but its communication heavily relies on the model dimension. DeComFL (Li et al., 2025b) pioneeringly exploited the intrinsic properties of ZO gradients - specifically, their decomposition into gradient scalars and perturbation vectors determined by random seeds - to achieve dimension-free communication overhead in LLM fine-tuning. Yet, it suffers from slower convergence due to the nature of ZO-SGD.

# 3 A GENERALIZED SCALAR-ONLY COMMUNICATION IN FL FRAMEWORK

In this section, we will present a generalized FL framework with scalar-only communication. Before that, we make a brief review about the zeroth-order method and its application for the dimension-free communication in FL, which will be the two key pillars for the following algorithm design.

## 3.1 ZEROTH-ORDER SGD AND SCALAR REPRESENTATIONS

We focus on the randomized gradient estimator (RGE) for performing ZO gradient estimation in this paper. It is also commonly referred to as Simultaneous Perturbation Stochastic Approximation (SPSA) (Spall, 1992; Nesterov & Spokoiny, 2017). Given a scalar-valued loss function $f(x)$ where $x \in \mathbb{R}^d$, the forward-style RGE is

$$\hat{\nabla} f(x) = \frac{1}{\mu}\big(f(x + \mu u) - f(x)\big)u, \ u \sim \mathcal{N}(0, I_d), \quad (1)$$

where $u$ represents a random direction vector sampled from a standard Gaussian distribution and $\mu > 0$ is a small constant, commonly termed the smoothing parameter, controlling the perturbation step size.

Figure 1: An illustration of ZO update.

An intriguing attribute of RGE is its efficient representation using only two scalars. First, we introduce a gradient scalar $g := \frac{1}{\mu}(f(x + \mu u) - f(x)) \in \mathbb{R}$, which serves as a scaling constant capturing the directional derivative. $g$ can also be explained as an approximate value for the directional gradient. Second, due to the deterministic nature of pseudo-random number generators, the random direction vector $u \in \mathbb{R}^d$ can be uniquely determined by a random seed $s$. Hence, $\hat{\nabla} f(x)$ can be efficiently expressed by two scalars. Crucially, this compact representation significantly enhances the efficiency of model updates in ZO frameworks. To illustrate, consider ZO-SGD update rule shown in Fig. 1:

$$x_{R+1} = x_R - \frac{\eta}{\mu}\big(f(x_R + \mu u_R) - f(x_R)\big)u_R = x_R - \eta g_R u_R = \cdots = x_0 - \eta \sum_{r=0}^{R} g_r u_r \quad (2)$$

This implies that, given the initial point $x_0$, a few number of gradient scalars $\{g_r\}$ and random seeds $\{s_r\}$ are sufficient to reconstruct $x_R$, irrespective of the dimensionality $d$ of $x$. This representation will play a crucial role in the dimension-free communication FL algorithm that follows.

## 3.2 FEDERATED LEARNING WITH DIMENSION-FREE COMMUNICATION

We consider a FL scenario with $M$ clients, each owning a local loss function $f_i$. The goal is to collaboratively minimize the global loss function across all clients without sharing their private data:

$$\min_{\boldsymbol{x} \in \mathbb{R}^d} f(\boldsymbol{x}) = \min_{\boldsymbol{x} \in \mathbb{R}^d} \frac{1}{M} \sum_{i=1}^{M} f_i(\boldsymbol{x}), \quad \text{where } f_i(\boldsymbol{x}) := \mathbb{E}\left[F_i(\boldsymbol{x}; \xi_i)\right]. \quad (3)$$

A typical FL round consists of two communications: 1) **Downlink Communication**: The server broadcasts the current aggregated global model to a subset of clients; 2) **Uplink Communication**: The selected clients return their locally updated model to the server. Both can be an expansive communication operation when the number of parameters $d$ is large.

The core idea of dimension-free communication in FL is leveraging the scalar representation of ZO-SGD to avoid transmitting the full models. To illustrate that, consider the following global model update rule with the notation that $x_{r,\tau}^{(i)}$ denotes client $i$'s model at the $r$-th round and $\tau$-th local update step and $x_r$ denotes the $r$-th global model:

$$x_{r+1} = \frac{1}{|C_r|} \sum_{i \in C_r} x_{r,\tau}^{(i)} = x_r + \frac{1}{|C_r|} \sum_{i \in C_r} (x_{r,\tau}^{(i)} - x_r) = x_r - \eta \frac{1}{|C_r|} \sum_{i \in C_r} \sum_{k=0}^{\tau-1} g_{r,k}^{(i)} u_{r,k}, \quad (4)$$

where $C_r$ is the set of sampled clients in the $r$-th round, $u_{r,k}$ are generated by shared random seeds across all clients, ensuring that all clients move along consistent directions. It enables that the global aggregation step in the server is simply computing an average of the gradient scalars: $g_{r,k} = \frac{1}{|C_r|} \sum_{i \in C_r} g_{r,k}^{(i)}$ from the local gradient scalar $g_{r,k}^{(i)} = \big(f_i(x_{r,k}^{(i)} + \mu u_{r,k}) - f_i(x_{r,k}^{(i)})\big)/\mu$.

**Uplink Communication:** From Eq. (4), sampled clients only transmit local gradient scalars $g_{r,k}^{(i)}$ to the server for global aggregation. **Downlink Communication:** ZO scalar representation only captures relative updates, so it is crucial to ensure that the server and all clients start from the same starting point. To achieve this, a model-reset mechanism is introduced: after finishing their local updates in each round, each participants resets its local model to the initial model, which is the global server model by induction. With this reset mechanism, the downlink communication can be conceptualized similarly to Eq. (4), with the distinction that clients may be absent in multiple rounds.

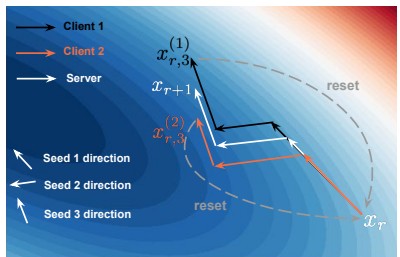

Figure 2: One-round update with 2 clients and 3 local updates. They share the same direction for each local update with different lengths. Arrive $x_{r+1}$ for both clients requires **7 steps**: 3 local updates, reset and 3 updates with global values.

Unlike the standard FL algorithm, reconstructing instead of pulling the model is used for catching the current server model through global gradient scalars and random seeds from preceding missed rounds. Consequently, the server is required to maintain auxiliary information: the client's last participation round, historical random seeds, and the global gradient scalars. This constitutes a negligible extra memory requirement as these are merely a few scalar values. We demonstrate the process in Fig. 2.

### 3.3 GENERALIZED SCALAR-ONLY COMMUNICATION IN FEDERATED LEARNING

In the work by Li et al. (2025b), the inherent dependency on ZO-SGD limits its applicability and the full potential of its dimension-free communication framework. One of our key contributions is observing that the crucial element is not the specific choice of ZO-SGD, but the basic use of scalar representations. Specifically, by maintaining records of their respective states with the update constructed by these scalar representations, the server and clients can effectively accommodate a wider range of optimization algorithms within the dimension-free communication paradigm. Thus, in Algorithm 1, we present a more generalized formulation that allows for the integration of various optimization techniques.

In Algorithm 1, communication is structured as follows: clients transmit $\{\Delta x_{r,k}^{(i)}\}_{k=1}^K$ to the server for global aggregation, and the server distributes the aggregated update $\Delta x_r$ to clients for model reconstruction. The dimension-independent property is preserved if both client-side updates $\Delta x_{r,k}^{(i)}$ and the server-side aggregated update $\Delta x_r$ can be effectively represented by scalars. Note a persistent state may be required to reconstruct $\Delta x_r$ with $r_l$ as the last participated round.

## 4 HESSIAN-INFORMED SCALAR-ONLY COMMUNICATION IN FL (HISO)

### 4.1 FIND A BETTER ASCENT $\Delta x_{r,k}^{(i)}$ DIRECTION

We use the proposed generalized framework to design a novel method superior to ZO-SGD-based FL while retaining dimension-free communication. A core challenge in the preceding framework is to

identify an effective ascent direction $\Delta x_{r,k}^{(i)}$ that is constructible solely from scalar values and current state information. While ZO-SGD meets these requirements, a superior alternative can be found.

Recall that the ZO methods' slow convergence is due to its dependency on random search directions (Ma & Huang, 2025). More specifically, recall the Eq. (1) with $u \sim \mathcal{N}(0, I)$, which uniformly searches all directions in the $\mathbb{R}^d$ space, is the update direction regardless of the scalar $g$. A natural extension is that we can guide the search direction with an invertible matrix $H_r$. Suppose $H_r$ is given, the Line 11 in Algo. 1 can be formulated as the following sub-optimization problem

$$\min_{g \in \mathbb{R}} \ \|\nabla f_i(x_{r,k}^{(i)}) - \Delta x_{r,k}^{(i)}\|_2^2 \qquad \text{(Ascent Direction)} \quad (5)$$

$$\text{s.t.} \ \ \Delta x_r^{(i)} = g \cdot H_r^{-1/2} u_{r,k}, \ \ u_{r,k} \sim \mathcal{N}(0, I_d) \in \mathbb{R}^{d \times 1} \qquad \text{(Scalars Representation)} \quad (6)$$

It will be clear later why we use this strange $H_r^{-1/2}$ notation instead of $H_r$ directly. Solving the above least-squares problem, we have

$$g^o = (u_{r,k}^\mathsf{T} H_r^{-1} u_{r,k})^{-1} u_{r,k}^\mathsf{T} H_r^{-1/2} \nabla f_i(x_{r,k}^{(i)}) \qquad (7)$$

Note $(u^\mathsf{T} H^{-1} u)^{-1}$ is a scalar that is independent of iterates $x_{r,k}^{(i)}$. Hence, we can absorb it into the learning rate. Next, note that $u_{r,k}^\mathsf{T} H_r^{-1/2} \nabla f_i(x_{r,k}^{(i)}) = \frac{1}{\mu}\big(f_i(x_{r,k}^{(i)} + \mu H_r^{-1/2} u_{r,k}) - f_i(x_{r,k}^{(i)})\big) + \mathcal{O}(\mu)$. Hence, we obtain the following update rule

$$\boxed{\Delta x_{r,k}^{(i)} = \frac{1}{\mu}\big(f_i(x_{r,k}^{(i)} + \mu H_r^{-1/2} u_{r,k}) - f_i(x_{r,k}^{(i)})\big) H_r^{-1/2} u_{r,k}} \qquad (8)$$

Now it should be clear why we use the notation $H_r^{-1/2}$ after we take the expectation of $\Delta x_{r,k}^{(i)}$:

$$\mathbb{E}\, \Delta x_{r,k}^{(i)} \approx \mathbb{E}\, H_r^{-1/2} u_{r,k} u_{r,k}^\mathsf{T} H_r^{-1/2} \nabla f_i(x_{r,k}^{(i)}) = H_r^{-1} \nabla f_i(x_{r,k}^{(i)}). \qquad (9)$$

When $H_r$ is a well-approximated Hessian matrix, the expectation of gradient descent follows the Newton-style gradient descent (Boyd & Vandenberghe, 2004). The first-order counterpart of $\Delta x_{r,k}^{(i)}$ is called natural gradient since it can be viewed as a pre-conditioned gradient (Amari, 1998). Recalling the linear transformation property of Gaussian Distribution, the update Eq. 8 can be more concisely written as the following form

$$\Delta x_{r,k}^{(i)} = \frac{1}{\mu}[f_i(x_{r,k}^{(i)} + \mu z_{r,k}) - f_i(x_{r,k}^{(i)})]z_{r,k}, \ \ z_{r,k} \sim \mathcal{N}(0, H_r^{-1}). \qquad (10)$$

This formulation also aligns with recent work by Ye et al. (2018) and Zhao et al. (2025), which refers to this type of update as Hessian-Informed or Hessian-Aware Zeroth-Order Optimization.

## 4.2 LEARNING GLOBAL CURVATURE WITHOUT EXTRA COMMUNICATION COST

A follow-up question for the above formulation is how to find this $H_r$ matrix. One plausible approach is, again, utilizing the zeroth-order gradient estimators to approximate directional second derivatives

$$u^\mathsf{T} \nabla^2 F(x) u \approx \frac{F(x + \mu u) + F(x - \mu u) - 2F(x)}{2\mu^2}, \ \ u \sim \mathcal{N}(0, I_d). \qquad (11)$$

However, this approach has two limitations: 1) this requires an additional function evaluation per direction and extra communications; 2) forming the full $d \times d$ Hessian is both costly and unnecessary. Instead, we only seek a diagonal preconditioner, akin to Adam's per-coordinate scaling (Kingma & Ba, 2015)[2]. Recall the global update term $\Delta x_{r,k}$ approximates the value of the gradient and it can be constructed by scalars only as discussed before. Further, notice this value is needed for the reconstruction step. Hence, we have a free variable to approximate the diagonal Hessian through the following proposed rule. We only update the Hessian at the beginning of one communication round with $\tau$-local update steps followed by the exponential moving averaging:

$$H_{r+1} = H_{r,\tau} = (1-\nu)H_{r,\tau-1} + \nu \frac{1}{m} \sum_{i \in S_r} \text{Diag}([\Delta x_{r,\tau}]^2 + \epsilon I)$$
$$\vdots$$

---

[2]More accurately, our method resembles RMSProp as it currently is without a momentum term. Momentum could be incorporated without additional communication costs using the same technique presented in this section. Given the existing length of this paper, we will not elaborate on this momentum extension here.

---

**Algorithm 1** Generalized Scalar-only Communication in Federated Learning

---

1: **Initialize**: learning rate $\eta$, local update steps $\tau$, communication rounds $R$.
2: **Allocate**: memory for recording the necessary historical states and client's participation information.
3: **for** $r = 0, \cdots, R - 1$ **do**
4:     Server uniformly samples a client set $C_r$ and distributes the shared random seeds $\{s_r\}$.
5:     **for** each client $i \in C_r$ **in parallel do**
6:         Receive the necessary scalar representations of $\{\Delta x_{r'}\}$ from server.
7:         Reconstruct the $\{\Delta x_{r'}\}$ from the scalars and update state.
8:         $x_{r,0}^{(i)} = x_{r_l,\tau}^{(i)} - \eta \sum_{r'=r_l}^{r-1} \Delta x_{r'}$                   $\triangleright$ Equivalent to pull model
9:         **for** $k = 0, \cdots, \tau - 1$ **do**
10:            Find $\Delta x_{r,k}^{(i)}$ that 1) is ascent direction; 2) can be represented by scalars + state;
11:            $x_{r,k+1}^{(i)} = x_{r,k}^{(i)} - \eta \Delta x_{r,k}^{(i)}$.             $\triangleright$ Client local update
12:         **end for**
13:         $x_{r,\tau}^{(i)} \Leftarrow x_{r,0}^{(i)}$ reset the model and other necessary states.
14:         Send the necessary scalar representations of $\{\Delta x_{r,k}^{(i)}\}$ to server.    $\triangleright$ Equivalent to push model
15:     **end for**
16:     Aggregate the scalar representations of $\{\Delta x_{r,k}^{(i)}\}$ into the ones for the global $\Delta x_r$.
17: **end for**

---

$$H_{r,1} = (1 - \nu)H_r + \nu \frac{1}{m} \sum_{i \in S_r} \text{Diag}([\Delta x_{r,0}]^2 + \epsilon I), \tag{12}$$

where $\epsilon$ is a small number to make sure that $H_{r+1}$ is strictly positive definite.

This Adam-style method, similar to its first-order counterpart (Reddi et al., 2021), has two advantages: 1) the diagonal matrix approximation avoids the $d^2$ storage for the Hessian matrix, making the proposed method scalable with the large-scale model. 2) the vector $\Delta x_{r,k}$ can be represented by scalars, so the server and clients can reconstruct this global Hessian without any extra communication.

## 4.3 PUTTING TOGETHER TO ESTABLISH THE DESIGN OF **HiSo**

**HiSo** is established by substituting the previously determined ascent direction and the global Hessian learning method into the scalar-only communication framework. A illustration of **HiSo** is shown in Fig. 3. **To elucidate the basic HiSo with brevity, we write out a simplified case where one local update occurs per round ($\tau = 1$). The following equation is for one round update of one client.**

$$
\begin{aligned}
&\text{for } t = r_l, \cdots r - 1: \\
&\quad \Delta x_t = g_t H_t^{-1/2} u_t, \quad u_t \Leftarrow \mathcal{N}(\text{seed}_t) \\
&\quad x_{t+1}^{(i)} = x_t^{(i)} - \eta \Delta x_t \\
&\quad H_{t+1} = (1 - v)H_t + \nu \text{Diag}([\Delta x_t]^2 + \epsilon I)
\end{aligned}
\left.\vphantom{\begin{aligned}&1\\&1\\&1\\&1\end{aligned}}\right\} \text{(Reconstruct States for the Missing Rounds)}
$$

$$
\begin{aligned}
&\Delta x_r^{(i)} = \frac{1}{\mu}[f_i(x_r^{(i)} + \mu H_r^{-1/2} u_r) - f_i(x_r^{(i)})] H_r^{-1/2} u_r \\
&x_{r+1}^{(i)} = x_r - \eta \Delta x_r^{(i)} \\
&x_{r+1}^{(i)} \Leftarrow x_r \quad (\text{reset})
\end{aligned}
\left.\vphantom{\begin{aligned}&1\\&1\\&1\end{aligned}}\right\} \text{(Client Local Update)}
$$

$$
\Delta x_r = \frac{1}{|C_r|} \sum_{i \in C_r} \Delta x_r^{(i)} = \left(\frac{1}{|C_r|} \sum_{i \in C_r} g_r^{(i)}\right) H_r^{-1/2} u_r
\left.\vphantom{\sum_{i \in C_r}}\right\} \text{(Global Aggregation at Server)}
$$

where $r_l$ is the last participated round, $x_r^{(i)}$ is $i$-th client's model at communication round $r$ and we omit the $k$ for local-update while $x_r$ is the global/server model. The same notation conventions apply for $g_r^{(i)}$, $g_r$, $\Delta x_r^{(i)}$ and $\Delta x_r$. Though mathematically equivalent, this representation is presented by disregarding implementation and communication intricacies to highlight the core mechanics better. Nevertheless, it is essential to highlight that only $g_r^{(i)}$, $g_r$ and random seeds are required to be

communicated between clients and server as our scalar-only framework proposes. For the detailed algorithm table with all features, we provide it in the Appendix D.

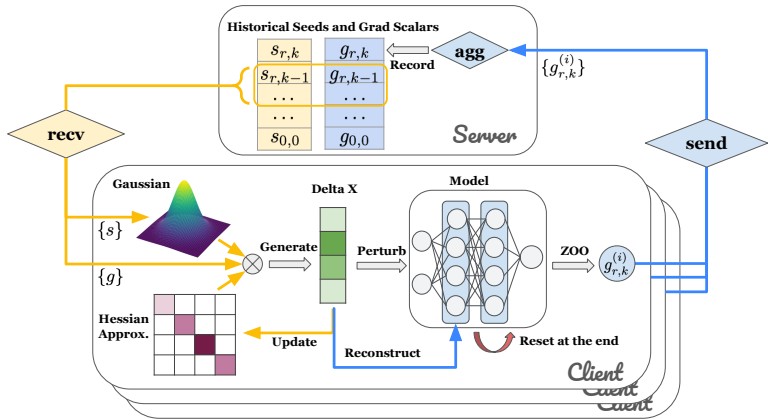

Figure 3: Illustration of `HiSo`

## 5 PERFORMANCE ANALYSIS

### 5.1 HESSIAN, VARIANCE OF ZO GRADIENT, AND LOW EFFECTIVE RANK ASSUMPTION

To lay the foundation for analyzing `HiSo`, we first examine a basic component of ZO: the estimation of the variance term. It provides essential insights into Hessian-informed ZO methods.

$$\mathbb{E}\|u\|_\Sigma^2 := \mathbb{E}\, u^\mathsf{T}\Sigma u, \quad u \sim \mathcal{N}(0, I_d) \in \mathbb{R}^{d\times 1}, \tag{13}$$

where $\Sigma$ is some semi-positive Hessian matrices[3]. The standard $L$-smoothness assumption implies that $\|\Sigma\| \le L$. Consequently, the preceding quantity can be upper-bounded as:

$$\mathbb{E}\|u\|_\Sigma^2 \le \|\Sigma\| \cdot \mathbb{E}\|u\|^2 \le Ld, \tag{14}$$

Note that the upper bound derived above can be quite large if the dimension $d$ is large. This dependence on dimensionality is a well-known factor leading to a typically slow convergence rate of ZO methods (Nesterov, 2013). Fortunately, this bound only represents a worst-case scenario. Motivated by empirical observations that the Hessian of trained large language models (LLMs) possesses relatively few eigenvalues significantly far from zero (Papyan, 2020; Yao et al., 2020; Wu et al., 2020), Malladi et al. (2023) proposed a low-effective rank assumption. This spectral property, where most eigenvalues are concentrated near zero, is illustrated in Fig. 4 (left). To utilize this assumption, we need to treat the variance more carefully:

$$\mathbb{E}\|u\|_\Sigma^2 = \mathrm{Tr}(\Sigma\mathbb{E}\, uu^\mathsf{T}) = L\,\mathrm{Tr}(\Sigma/L) := L\kappa, \tag{15}$$

where $\kappa = \mathrm{Tr}(\Sigma/L)$ is called the effective rank of Hessian $\Sigma$. It is computationally prohibitive to find the exact value of $\kappa$, but several previous workers indicate $\kappa \ll d$ (Li et al., 2025b; Malladi et al., 2023). Hence, we get a tighter variance estimation. Utilizing the Hessian approximate matrix, we can further improve this bound. Supposing we have a well approximation matrix $H$ for the Hessian $\Sigma$, the weighted Gaussian vector $z$ is sampled from the distribution $\mathcal{N}(0, H^{-1})$. Then, we have

$$\mathbb{E}\|z\|_\Sigma^2 = \mathbb{E}\,\mathrm{Tr}(H^{-1/2}\Sigma H^{-1/2}uu^\mathsf{T}) = \mathrm{Tr}(H^{-1/2}\Sigma H^{-1/2}) := \zeta, \tag{16}$$

where we call the quantity $\zeta$ as the low whitening rank of Hessian $\Sigma$.

If $H$ is the perfect approximation of $\Sigma$, then $\zeta = d$. This case is neither possible in practice nor ideal in LLM cases. Recalling that only a few eigenvalues of $\Sigma$ are non-zero, then $H \approx \mathrm{Diag}(\Sigma + \epsilon\mathbb{1})$ is a more effective inverse value, which is similar to Wiener filtering in the denoising field (Sayed, 2003). Now we summarize the above discussion into the following definition.

| Assumption | $\mathbb{E}\|u\|_\Sigma^2$ | $\mathbb{E}\|z\|_\Sigma^2$ |
|---|---|---|
| $L$-smooth | $Ld$ | $2d$ |
| Low Effective Rank | $L\kappa$ | $\zeta$ |

Table 1: The Upper-Bound of ZO Gradient Variance

---

[3]For a non-convex function, Hessian may contain some negative eigenvalues. One possible choice of $\Sigma$ can be the absolute eigenvalues of the Hessian.

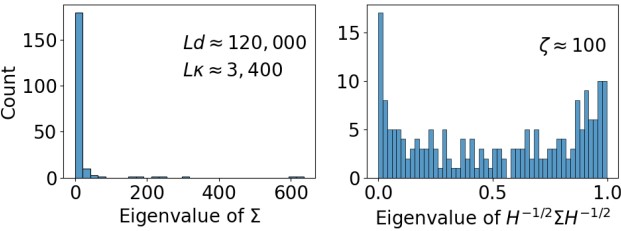

Figure 4: An Illustration of the Eigenvalue Distribution.

**Definition.** We call a diagonal matrix $H$ as **a well-approximate matrix of Hessian** $\Sigma$ if the whitening matrix $\Xi := H^{-1/2}\Sigma H^{-1/2}$ satisfies the following condition:

$$\text{Tr}(\Xi) = \text{Tr}(H^{-1/2}\Sigma H^{-1/2}) \leq \begin{cases} 2d & (L\text{-Smoothness}) \\ \zeta & (\text{Low Effective Rank}) \end{cases}, \tag{17}$$

where $\zeta$ is a quantity independent of the dimension $d$, and the factor 2 is just a safety factor to tolerate the imperfect inverse. The above assumptions and results are summarized in Table 1.

To illustrate the effectiveness of this whitening process, we execute a simple numerical experiment. To simulate the distribution of Hessian eigenvalues, we assume that there are 200 eigenvalues following the log-normal distribution, i.e., $\log(\Sigma) \sim \mathcal{N}(0, 3I)$. The simulation, depicted in Fig. 4, shows that $\zeta \ll L\kappa \ll Ld$. This lays the theoretical foundation for the acceleration of our proposed `HiSo`.

### 5.2 CONVERGENCE RESULTS

We first present some standard assumptions that will be used to establish the convergence results.

**Assumption 1** (*L*-Lipschitz). *Suppose the global loss function $F$ is L-smooth, i.e., for all $x, y \in \mathbb{R}^d$, we have $\|\nabla F(x) - \nabla F(y)\| \leq L\|x - y\|$.*

**Assumption 2** (Unbiased Stochastic Gradients with Bounded Variance). *The stochastic gradient computed by clients is unbiased with bounded variance:* $\mathbb{E}[\nabla f_i(x; \xi)] = \nabla f_i(x)$ *and* $\mathbb{E}\|\nabla f_i(x; \xi) - \nabla f_i(x)\|^2 \leq \sigma_s^2, \forall x$, *where $\xi$ represents a data sample.*

**Assumption 3** (Bounded Heterogeneity). *The cost function satisfies $\|\nabla f_i(x) - \nabla F(x)\| \leq \sigma_G, \forall x$.*

**Assumption 4** (Bounded Learned Hessian). *The learned Hessian has $0 < \beta_\ell \leq \|H_r\| \leq \beta_u, \forall r$.*

The last assumption is common in Hessian-informed (Maritan et al., 2024; Zhao et al., 2025) or Adam-style algorithms (Kingma & Ba, 2015; Reddi et al., 2021), where the requirement of bounded gradient implies this assumption directly. It is worth noting that, unlike the assumption on Hessian, the parameters $\beta_\ell$ and $\beta_u$ can be easily controlled in the algorithm design by adding the clipping step (Liu et al., 2024). This assumption also implies $\beta_u^{-1} \leq \|H_k^{-1}\| \leq \beta_\ell^{-1}$.

**Theorem 1.** *Under Assumptions 1, 2, 3, and 4, if $\eta \leq \min\left(\frac{\beta_\ell}{mL}, \frac{1}{8\rho_k}, \frac{\beta_\ell}{4(\tau-1)}\sqrt{\frac{1}{L(d+2)}}\right)$ and denote $\Delta_{1,*} := F(\bar{x}_1) - F^\star$, the sequence of iterates generated by `HiSo` satisfies:*

$$\frac{1}{\tau R}\sum_{r=0}^{R-1}\sum_{k=0}^{\tau-1}\mathbb{E}\|\nabla F(\bar{x}_{r,k})\|_{H_r^{-1}}^2 \leq \frac{4\Delta_{1,*}}{\eta\tau R} + \underbrace{\frac{32\eta(\tau-1)^2 L\bar{\phi}}{\beta_\ell\tau m}(\sigma_G^2 + \sigma_s^2)}_{\text{extra client drift term}} + \frac{16\eta\bar{\rho}}{\beta_\ell m}(\sigma_G^2 + \sigma_s^2) + O(\eta\mu),$$

*where $\bar{x}_{r,k} = \frac{1}{M}\sum_{i=1}^{M} x_{r,k}^{(i)}$, $\bar{\rho} = \frac{1}{\tau R}\sum_r\sum_k(\text{Tr}(H_r^{-1/2}\Sigma_{r,k}H_r^{-1/2}) + 2\|H_r^{-1/2}\Sigma_{r,k}H_r^{-1/2}\|)$, $\Sigma_{r,k}$ is a PSD matrix that upper-bounds Hessian at $x_{r,k}$ and $\bar{\phi} = \frac{1}{R}\sum_r(\text{Tr}(H_r^{-1}) + 2\|H_r^{-1}\|)$.* ∎

Roughly, $\bar{\rho}$ can be understood as the sum of whitening Hessian eigenvalues and $\bar{\phi}$ as the sum of approximate Hessian eigenvalues. $\bar{\rho}$ consist of two parts: 1) $\text{Tr}(H_r^{-1/2}\Sigma_{r,k}H_r^{-1/2})$ is the quantity discussed previously, 2) $\|H_r^{-1/2}\Sigma_{r,k}H_r^{-1/2}\|$, typically, is much smaller than the first term when the model dimension $d$ is large. The properties of the terms in $\bar{\phi}$ are similar to $\bar{\rho}$.

**Corollary 1** (Convergence Rate for `HiSo`). Suppose the learned global Hessian $H_r$ satisfies the well-approximated condition (17). When $\tau = 1$ and $\eta = \sqrt{m\beta_\ell/\bar{\rho}R}$, `HiSo`'s convergence rate is

$\mathcal{O}(\sqrt{d/mR})$. Further, if the Hessian exhibits the low-effective rank property, the rate can be further improved to $\mathcal{O}(\sqrt{\zeta/mR})$ independent of the model dimension $d$ and the Lipschitz constant $L$.

**Corollary 2** (Convergence Rate for DeComFL). Note that DeComFL (Li et al., 2025b) can be regarded as a special case of `HiSo` with $H_r \equiv I, \forall r$ and $\beta_\ell = \beta_u = 1$. Therefore, we can immediately recover the convergence rate of DeComFL with $\tau = 1$ is $\mathcal{O}(\sqrt{Ld/mR})$ with standard assumptions or $\mathcal{O}(\sqrt{L\kappa/mR})$ with the extra low-effective rank phenomenon.

**Corollary 3** (Convergence Rate for $\tau > 1$ case). When the local update step $\tau > 1$, the difference between `HiSo` and DeComFL becomes bigger. Under the well-approximate and low whitening rank scenario, the convergence rate of `HiSo` is $\mathcal{O}(\sqrt{\zeta/\tau mR}) + \mathcal{O}(\sqrt{\tau\kappa/mR})$, still independent of the model dimension $d$ and Lipschitz condition $L$; meanwhile, DeComFL becomes dependent on $d$ again. This resolved the previous open question that DeComFL (Li et al., 2025b) cannot provide the convergence rate with a low-effective rank assumption when $\tau > 1$. See Appendix F.6.2 for details.

**Remarks about well-approximated condition.** It is important to note that Theorem 1 does not require the well-approximated condition. This condition is only necessary for the three preceding corollaries, which utilize it to establish clean convergence rates. Although it is hard to determine if this approximation holds in the context of LLMs, the assumption offers a plausible explanation for the rapid convergence often observed in practice (where the required iterations are much smaller than $d$). If $H_r$ fails to yield an effective Hessian approximation, the performance of `HiSo`, at worst case, degenerates into DeComFL. We provide more discussion on this point in Appendix F.7.1.

## 6 Experiments

**The Global Diagonal Hessian Approximation $H$.** We begin by training a simple CNN model on MNIST (LeCun et al., 1998) to visualize the learned diagonal Hessian approximation $H$. We set up a 64-client FL environment where data was partitioned non-IID using a Dirichlet distribution ($\alpha = 1$). Each communication round involved randomly sampling 8 clients for training. Evaluating the Hessian smoothing parameter $\nu$ revealed negligible impact on convergence and final accuracy (Fig. 5, left), demonstrating the algorithm's robustness to this hyperparameter. Furthermore, Fig. 5 (right) plots each entry of the learned diagonal Hessian values at the end of training. While individual entries may appear stochastic, their overall distribution clearly exhibits a long-tail phenomenon. This observation aligns with the low effective rank assumption discussed in Sec. 5.1. Although computing the exact Hessian is computationally prohibitive, the rapid convergence combined with this observed distribution suggests our strategy effectively approximates relevant Hessian structure. We design more experiments and provide more direct evidences in Appendix F.7.2.

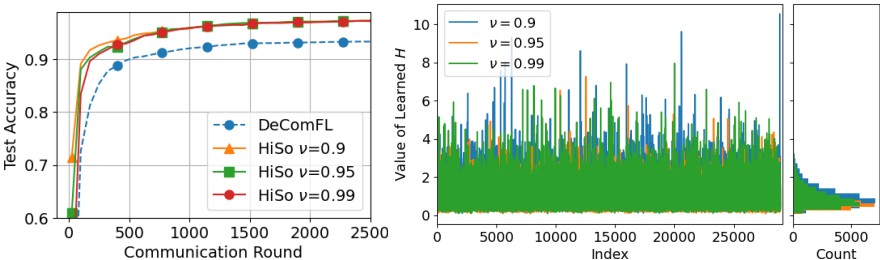

Figure 5: Ablation study of smoothing parameter $\nu$ and the distribution of the learned global Hessian $H$.

**`HiSo` is Faster Than DeComFL in Small Model Training Tasks.** In Fig. 5, we evaluate `HiSo` against another dimension-free communication FL method - DeComFL. Crucially, the communication cost per round are the same for both to ensure a fair comparison of algorithmic efficiency. Fig. 5 shows that, under the same communication constraints, our `HiSo` achieves significantly faster convergence and reaches a superior final performance level compared to DeComFL. For this comparison, both were tuned using their optimal learning rates. More comparison is provided in Appendix E.

**LLM Fine-Tuning Task Setup:** Our FL system consists of 6 clients in total, and 2 clients are uniformly sampled in each round. To comprehensively evaluate `HiSo`'s performance, we execute sentiment classification on SST-2 (Socher et al., 2013), question matching on QQP, and question answering on SQuAD (Rajpurkar et al., 2016). We set $P = 5$ for all ZO methods.

**`HiSo` Accelerates Training with Less Communication Cost in LLM Fine-Tuning.** In Table 2, `HiSo` consistently reduces communication rounds required to reach DeComFL's best test accuracy, resulting in lower communication cost. Specifically, `HiSo` achieves 1.4-5.4× speedup with 29%-80% communication savings across all tasks. These results show that `HiSo` accelerates convergence and reduces communication cost, making it more practical for large-scale FL scenarios involving LLMs.

Table 2: `HiSo`'s Acceleration. For DeComFL, we report the total number of communication rounds required to fully converge. For `HiSo`, we report the number of rounds needed to match DeComFL's best test accuracy, along with the corresponding communication cost.

| Model | Method | SST-2 | | | QQP | | | SQuAD | | |
|-------|--------|-------|---------|-----------|------|---------|-----------|-------|---------|-----------|
| | | Round | Speedup | Comm. Cost | Round | Speedup | Comm. Cost | Round | Speedup | Comm. Cost |
| OPT-350M | DeComFL | 550 | 1× | 21.56 KB | 775 | 1× | 30.35 KB | 1350 | 1× | 52.73 KB |
| | **`HiSo`** | 275 | 2× | 10.78 KB | 425 | 1.8× | 16.64 KB | 250 | 5.4× | 9.77 KB |
| OPT-1.3B | DeComFL | 1500 | 1× | 58.59 KB | 1125 | 1× | 43.95 KB | 350 | 1× | 13.67 KB |
| | **`HiSo`** | 1075 | 1.4× | 41.85 KB | 750 | 1.5× | 29.30 KB | 175 | 2× | 6.84 KB |
| OPT-2.7B | DeComFL | 1250 | 1× | 41.58 KB | 1475 | 1× | 48.75 KB | 450 | 1× | 15.65 KB |
| | **`HiSo`** | 775 | 1.6× | 26.21 KB | 975 | 1.5× | 32.11 KB | 200 | 2.3× | 6.94 KB |

**Extensive Baseline Comparison on LLM Fine-Tuning Tasks.** In Table 3, first-order methods (e.g., FedAvg, FedAdam, FedYogi and FedAdagrad) consistently achieve high test accuracy, but at the cost of TB-level communication volumes, which is quite challenging for real-world federated fine-tuning. As for ZO baselines, FedZO's communication cost is also quite high due to transmitting $d$-dimensional update. Although DeComFL achieves several orders of magnitude communication reduction, its cost is still higher than `HiSo` as it suffers from more rounds due to extremely slow convergence. Our proposed `HiSo` not only maintains the lowest communication cost in almost all tasks (only a little higher than DeComFL on OPT-1.3B+QQP) but also consistently outperforms ZO baselines in test accuracy.

Table 3: Performance for LLM Fine-Tuning. 1) We report the total communication cost of the single client during the entire training process until convergence, test accuracy for SST-2 and QQP, F1 score for SQuAD.

| Model | Method | SST-2 | QQP | SQuAD |
|-------|--------|-------|-----|-------|
| OPT-125M | FedAvg | $87.63\% \pm 0.16$ (0.15 TB) | $61.21\% \pm 0.37$ (0.08 TB) | $37.27 \pm 0.11$ (0.05 TB) |
| | FedAdam | $88.29\% \pm 0.47$ (0.30 TB) | $63.18\% \pm 0.31$ (0.06 TB) | $37.98 \pm 0.20$ (0.03 TB) |
| | FedYogi | $88.06\% \pm 0.33$ (0.29 TB) | $62.88\% \pm 0.21$ (0.05 TB) | $37.66 \pm 0.18$ (0.04 TB) |
| | FedAdagrad | $85.04\% \pm 0.51$ (0.18 TB) | $61.77\% \pm 0.22$ (0.06 TB) | $37.29 \pm 0.27$ (0.04 TB) |
| | FedZO | $84.19\% \pm 0.22$ (0.63 TB) | $60.06\% \pm 0.21$ (1.94 TB) | $34.03 \pm 0.26$ (0.14 TB) |
| | DeComFL | $85.21\% \pm 0.27$ (22.92 KB) | $60.11\% \pm 0.19$ (32.17 KB) | $34.12 \pm 0.22$ (17.42 KB) |
| | **`HiSo` (Ours)** | $85.55\% \pm 0.21$ (14.69 KB) | $60.72\% \pm 0.25$ (21.23 KB) | $35.26 \pm 0.14$ (7.12 KB) |
| OPT-350M | FedAvg | $89.79\% \pm 0.05$ (0.58 TB) | $63.32\% \pm 0.13$ (0.31 TB) | $43.38 \pm 0.13$ (0.12 TB) |
| | FedAdam | $89.92\% \pm 0.20$ (0.21 TB) | $63.28\% \pm 0.19$ (0.28 TB) | $45.92 \pm 0.14$ (0.08 TB) |
| | FedYogi | $89.68\% \pm 0.29$ (0.25 TB) | $63.21\% \pm 0.16$ (0.28 TB) | $45.01 \pm 0.25$ (0.09 TB) |
| | FedAdagrad | $87.42\% \pm 0.09$ (0.23 TB) | $62.55\% \pm 0.14$ (0.29 TB) | $44.49 \pm 0.11$ (0.09 TB) |
| | FedZO | $86.55\% \pm 0.23$ (0.68 TB) | $61.22\% \pm 0.30$ (0.66 TB) | $38.14 \pm 0.24$ (0.38 TB) |
| | DeComFL | $86.72\% \pm 0.28$ (21.56 KB) | $60.58\% \pm 0.16$ (30.35 KB) | $38.20 \pm 0.15$ (52.73 KB) |
| | **`HiSo` (Ours)** | $87.50\% \pm 0.22$ (17.33 KB) | $62.49\% \pm 0.17$ (18.63 KB) | $39.13 \pm 0.11$ (20.51 KB) |
| OPT-1.3B | FedAvg | $90.48\% \pm 0.35$ (0.63 TB) | $65.77\% \pm 0.20$ (0.32 TB) | $60.39 \pm 0.27$ (0.41 TB) |
| | FedAdam | $92.86\% \pm 0.43$ (0.79 TB) | $64.59\% \pm 0.53$ (1.10 TB) | $61.56 \pm 0.14$ (0.27 TB) |
| | FedYogi | $92.39\% \pm 0.58$ (0.83 TB) | $64.44\% \pm 0.22$ (1.12 TB) | $61.44 \pm 0.19$ (0.29 TB) |
| | FedAdagrad | $90.92\% \pm 0.74$ (0.88 TB) | $64.05\% \pm 0.13$ (1.08 TB) | $60.72 \pm 0.23$ (0.33 TB) |
| | FedZO | $90.01\% \pm 0.29$ (4.73 TB) | $62.91\% \pm 0.14$ (3.53 TB) | $57.26 \pm 0.17$ (1.10 TB) |
| | DeComFL | $90.22\% \pm 0.10$ (58.59 KB) | $63.25\% \pm 0.11$ (43.95 KB) | $57.14 \pm 0.14$ (13.67 KB) |
| | **`HiSo` (Ours)** | $90.34\% \pm 0.12$ (49.18 KB) | $64.20\% \pm 0.13$ (96.67 KB) | $57.58 \pm 0.07$ (7.81 KB) |

In Appendix E, we present additional experimental results, along with comparisons and analyses of memory cost, communication cost, computation time, and other FL+PEFT baselines.

ACKNOWLEDGMENT

This work is supported in part by RIT CHAI Faculty Seed Grant, NIH award R16GM159671 and NSF grant CNS-2112471. The content is solely the responsibility of the authors and does not necessarily represent the official views of the funding agencies.

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

APPENDIX

## A  Statement of LLM Usage

LLM tools were used to help with language refinement and stylistic improvements. After each use of the tool, we carefully reviewed and validated the correctness and appropriateness of the generated text to ensure accuracy and alignment with the intended meaning.

## B  Conclusion and Limitations

In conclusion, we first present a generalized FL framework that supports scalar-only communication in both uplink and downlink, enabling the integration of a broader class of optimization algorithms beyond vanilla ZO-SGD. Building on this foundation, we propose `HiSo`, a Hessian-informed federated optimization algorithm that leverages diagonal Hessian approximations to accelerate convergence while preserving scalar-only communication efficiency without the demand to transmit any second-order information. From a theoretical perspective, we introduce a novel variance characterization for Hessian-informed zeroth-order gradients under a low-effective-rank assumption. This allows us to establish a convergence rate that is independent of both model dimensionality and function smoothness in non-convex settings - a result not previously achieved by any ZO method in FL. Our analysis further generalizes the previous framework and extends its theoretical guarantees to support multiple local updates, a critical component in practical FL deployments. The analysis offers a plausible explanation for the observed phenomenon of ZO convergence being much faster than its worst case. Empirically, `HiSo` consistently outperforms existing baselines, delivering higher test accuracy, up to about $5\times$ faster convergence, and substantially lower communication overhead. These results demonstrate the practical viability and theoretical soundness of unifying curvature-informed optimization with scalar-only communication in federated fine-tuning.

**Limitations:** The proposed method is currently limited by its treatment of the loss function $f_i$ as a generic one, without considering model-specific module structures. This is in contrast to modern parameter-efficient fine-tuning (PEFT) methods that often exploit properties like low-rank decomposition (e.g., $W = AB^{\mathsf{T}}$, where $A \in \mathbb{R}^{k_1 \times r}$ and $B \in \mathbb{R}^{k_2 \times r}$ and $r \ll k_1, k_2$). It is important to note that this explicit low-rank decomposition is distinct from the 'low effective rank' of the Hessian discussed in this paper. Consequently, there is potential to further refine our approach by designing Hessian information specifically tailored for PEFT methods such as LoRA (Hu et al., 2022) or GaLore (Zhao et al., 2024).

## C  Comparison of Related Works

From Table 4, we observe that `HiSo` achieves the best convergence rate among all ZO-FL related works (e.g., FedZO, BAFFLE, DeComFL) and is the first work to provide the rigorous convergence proof under the low effective rank assumption and supporting the $\tau > 1$ case at the same time. Compared with first-order related works, `HiSo` achieves significant communication improvement.

## D  Detailed `HiSo` Algorithm Table

Although the algorithm listed in the main context is quite complicated, it is simple if we ignore the dimension-free communication property. Mathematically, `HiSo` is equivalent to the following standard FedAvg style update

$$x_{r,0}^{(i)} = x_r \qquad\qquad\qquad \text{(Receive Model)}$$
$$\text{for } k = 0, 1, \cdots, \tau - 1:$$
$$g_{r,k}^{(i)} = \frac{1}{\mu}\big(f_i(x_{r,k}^{(i)} + \mu H_r^{-1/2} u_{r,k}) - f_i(x_{r,k}^{(i)})\big)$$
$$x_{r,k+1}^{(i)} = x_{r,k}^{(i)} - \eta g_{r,k}^{(i)} H_r^{-1/2} u_{r,k} \qquad\qquad \text{(Local Update)}$$
$$x_{r+1} = \frac{1}{|C_r|} \sum_{i \in C_r} x_{r,\tau}^{(i)} \qquad\qquad \text{(Aggregate Model)}$$
$$H_{r+1} = (1 - \nu)H_r + \nu \text{Diag}([x_{r+1} - x_r] \odot [x_{r+1} - x_r] + \epsilon I)$$

With that as reference, we present the full algorithm table for `HiSo`.

---

**Algorithm 2** Concrete Scalar Representations Communication with States for Federated Learning

---

1: **Initialize**: learning rate $\eta$, local update steps $K$, communication rounds $R$
2: **Allocate**: memory for recording the necessary historical states, including historical gradient scalars $\{g\}$, corresponding random seeds $\{s\}$ and clients' last participation round $\{r'\}$, which are initialized as 0.
3:
4: **for** $r = 0, 1, \cdots, R-1$ **do**
5:     Server uniformly samples a client set $C_r$ with cardinality $m$.
6:     Server randomly samples a random seed set $\{s_{r,k}\}_{k=0}^{\tau-1}$ and broadcasts it to all sampled clients.
7:     **for** each client $i \in C_r$ **in parallel do**
8:         $\{\{\Delta x_t^{(i)}\}_{k=0}^{\tau-1}\}_{t=r'}^{r-1} = \text{Rebuild}(\{\{s_{t,k}^{(i)}\}_{k=0}^{\tau-1}\}_{t=r_i'}^{r-1}, \{\{g_{t,k}^{(i)}\}_{k=0}^{\tau-1}\}_{t=r_i'}^{r-1})$
9:         $x_{r,0}^{(i)} = x_{r',0}^{(i)} - \eta \sum_{t=r'}^{r-1} \sum_{k=0}^{\tau-1} \Delta x_{t,k}^{(i)}$
10:        $\{g_{r,k}^{(i)}\}_{k=0}^{\tau-1} = \text{LocalUpdate}(\{s_{r,k}\}_{k=0}^{\tau-1})$
11:        Send $\{g_{r,k}^{(i)}\}_{k=0}^{\tau-1}$ back to the server.
12:     **end for**
13:     $\{g_{r,k}\}_{k=0}^{\tau-1} = \left\{ \frac{1}{|\mathcal{C}_r|} \sum_{i \in \mathcal{C}_r} g_{r,k}^{(i)} \right\}_{k=0}^{\tau-1}$        ▶ Global gradient scalar aggregation
14:     $\{\Delta x_{r,k}\}_{k=0}^{\tau-1} = \left\{ g_{r,k} H_r^{-1/2} u_{r,k} \right\}_{k=0}^{\tau-1}$       ▶ Global $\Delta$ aggregation at server
15:     Store $\{g_{r,k}\}_{k=0}^{\tau-1}$ and $\{s_{r,k}\}_{k=0}^{\tau-1}$ and update the client's last participation round $r_i' = r$.
16:     $x_{r+1} = x_r - \eta \sum_{k=0}^{\tau-1} \Delta x_{r,k}$       ▶ (Optional) Global model update
17: **end for**

---

**Algorithm 2a** Receiving Step for Hessian-Informed ZO Gradient for $i$-th Client at $r$-th Round

---

1: **Function** Rebuild($\{\{s_{t,k}\}_{k=0}^{\tau-1}\}_{r=r'}^{r-1}, \{\{g_{t,k}\}_{k=0}^{\tau-1}\}_{r=r'}^{r-1}$):     ▶ $r'$ is last participation round
2:     **for** $t = r', \cdots, r-1$ **do**
3:         **for** $k = 0, \cdots, \tau-1$ **do**
4:             Utilize the random seed $s_{t,k}$ to produce $u_{t,k} \sim \mathcal{N}(\mathbf{0}, \mathbf{I})$
5:             $\Delta x_{t,k} = g_{t,k} H_t^{-1/2} u_{t,k}$
6:             $H_{t+1} = (1 - \nu) H_t + \nu \text{Diag}([\Delta x_{t,\tau}]^2 + \epsilon I)$
7:         **end for**
8:     **end for**
9:     **return** $\{\{\Delta x_{t,k}\}_{k=0}^{\tau-1}\}_{t=r'}^{r-1}$       ▶ For model reconstruction

---

**Algorithm 2b** Sending Step for Hessian-Informed ZO Gradient for $i$-th Client at $r$-th Round

---

1: **Function** LocalUpdate($\{s_{r,k}\}_{k=0}^{\tau-1}$):
2:     **for** $k = 0, \cdots, \tau-1$ **do**
3:         Utilize the random seed $s_{r,k}$ to produce $u_{r,k} \sim \mathcal{N}(\mathbf{0}, \mathbf{I})$
4:         $g_{r,k}^{(i)} = \frac{1}{\mu} \left[ f_i(x_{r,k}^{(i)} + \mu H_r^{-1/2} u_{r,k}) - f_i(x_{i,r}^{(i)}) \right]$     ▶ Compute ZO gradient scalar
5:         $\Delta x_{r,k}^{(i)} = g_{r,k}^{(i)} H_r^{-1/2} u_{r,k}$   ▶ Can be replaced by other representation methods of $\Delta x_{r,k}^{(i)}$
6:         $x_{r,k+1}^{(i)} = x_{r,k}^{(i)} - \eta \Delta x_{r,k}^{(i)}$       ▶ Update local model
7:     **end for**
8:     $x_{r,\tau}^{(i)} \Leftarrow x_{r,0}^{(i)}$       ▶ Reset the local model and update other necessary states
9:     **return** $\{g_{r,k}^{(i)}\}_{k=0}^{\tau-1}$

---

Table 4: Comparison of Related Work. $d$ is the model dimension. $m$ is the number of sampled clients per round. $P$ is the number of perturbations. $R$ is the number of rounds. $\zeta$ is the low whitening rank of Hessian. $\kappa$ is the low effective rank of the Hessian. "LER" means the low effective rank assumption. "DF" mean dimension-free. "Proof on $\tau > 1$?" means whether the algorithm provides theoretical convergence proof under multiple local updates $\tau > 1$.

| Methods | Convergence Rate | LER? | Uplink | Downlink | Proof on $\tau > 1$? |
|---|---|---|---|---|---|
| FedAvg (McMahan et al., 2017) | $\mathcal{O}\left(\sqrt{\frac{L}{mR\tau}}\right)$ | ✗ | ✗ DF | ✗ DF | ✓ |
| FedAdam (Reddi et al., 2021) | $\mathcal{O}\left(\sqrt{\frac{L}{mR\tau}}\right)$ | ✗ | ✗ DF | ✗ DF | ✓ |
| FedZO (Fang et al., 2022) | $\mathcal{O}\left(\sqrt{\frac{Ld}{mPR\tau}}\right)$ | ✗ | ✗ DF | ✗ DF | ✓ |
| BAFFLE (Feng et al., 2024) | $\mathcal{O}\left(\sqrt{\frac{Ld}{mPR\tau}}\right)$ | ✗ | ✓ DF | ✗ DF | ✓ |
| DeComFL (Li et al., 2025b) | $\mathcal{O}\left(\sqrt{\frac{L\kappa}{mPR}}\right)$ | ✓ | ✓ DF | ✓ DF | ✗ |
| **HiSo** (This paper) | $\mathcal{O}\left(\sqrt{\frac{\zeta}{mPR\tau}}\right)$ | ✓ | ✓ DF | ✓ DF | ✓ |

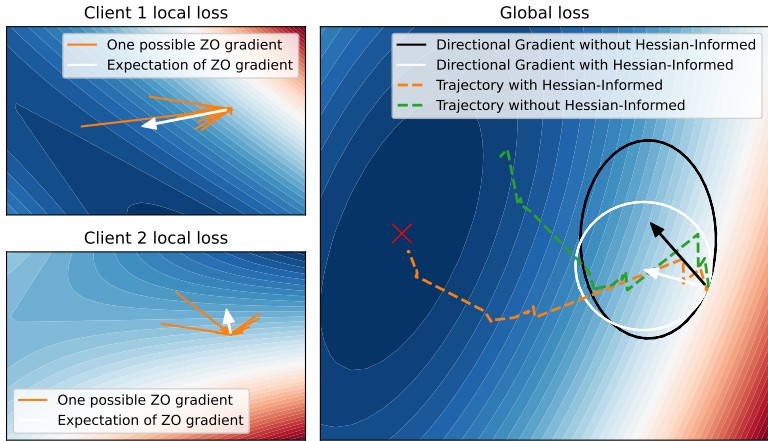

Client 1 local loss

Global loss

Client 2 local loss

Figure 6: An Illustration of Hessian-informed versus Regular ZO Gradient Direction under the FL Setting.

# E   EXTRA ANALYSIS, DISCUSSION, EXPERIMENT DETAIL AND RESULTS

## E.1   BASELINE SELECTION

To comprehensively evaluate **HiSo**'s performance, we select a broad range of classic or recent baselines covering both first-order and zeroth-order FL methods. We explain the reason why we choose those baselines as follows:

**First-Order FL Baselines:**

- **FedAvg**: the first and most classic first-order FL algorithm.
- **FedAdam**, **FedYogi** and **FedAdagrad**: adaptive gradient-based methods designed to accelerate convergence in FL.

**Zeroth-Order FL Baselines:**

- **FedZO**: the first FL method to incorporate ZO-SGD into client local updates.
- **DeComFL**: the first method to achieve dimension-free communication in FL, which also uses ZO-SGD to perform client local updates.

E.2 MEMORY COST RESULTS

In **HiSo**, there are several places that require additional memory to store extra information. We test their real memory consumption in our FL system.

(a) **Global Gradient Scalar and Seed** $\{g_r, s_r\}$ **Pairs**: In each communication round, only one $\{g_r, s_r\}$ pair is stored at the server, regardless of the number of clients. Specifically, storing two scalars per round over one million rounds would require only a few megabytes that is well within the capacity of any modern server.

(b) **Clients' Historical States (Last Participation Round)**: The server needs to store the last participation round (a scalar) for each client, which only consume a few megabytes for a FL system with even millions of clients. Specifically, storing 1 million client states (i.e., 1 million scalars in int 32) only needs 3.8 MB. Moreover, this storage cost can be optimized. For example, the server can transmit the last participation round index to each client, and the client can store it locally as a scalar. When the client returns, it simply sends this scalar back to server. This design eliminates the need for server to store all clients' states, so memory cost becomes negligible.

(c) **Clients' Hessian** $H$: We evaluate the peak memory usage of a single client across all baselines and our proposed **HiSo**. The results indicate that **HiSo** requires substantially less peak memory than all first-order baselines. Moreover, **HiSo** also outperforms FedZO in terms of memory efficiency, as in the original FedZO paper, it is not optimized for memory usage. Finally, when comparing **HiSo** with the memory-optimized DeComFL, we observe that **HiSo** still consumes less than twice the peak memory of DeComFL.

E.3 COMMUNICATION COST AND TRAINING ACCELERATION RESULTS

**HiSo is Extremely Communication-Efficient.** Fig. 7 shows the total communication cost of various FL methods across different model sizes (125M, 350M, and 1.3B), highlighting the dramatic efficiency of our **HiSo**. While traditional methods like FedAvg, FedZO, and FedAdam incur communication costs on the order of $10^{11}$ to $10^{13}$, **HiSo** reduces it by over 40 million times for 125M and 350M models, and up to 90 million times for the 1.3B model. Even compared to the strongest communication-efficient baseline DeComFL, **HiSo** still achieves noticeably lower communication cost because accelerated convergence introduces less training rounds. This substantial reduction shows that **HiSo** is highly communication-efficient and particularly well-suited for large-scale FL with high-capacity models. More experiment details are provided in Appendix E.

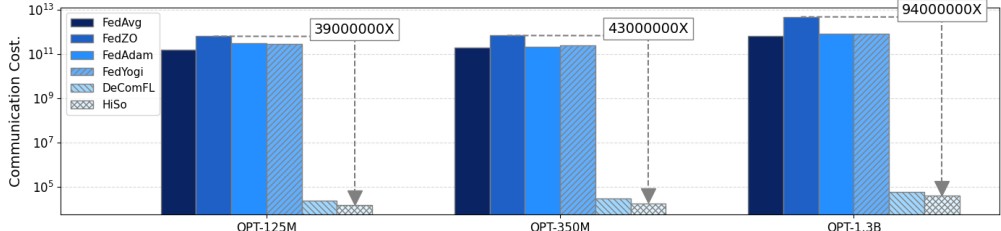

Figure 7: Communication Overhead Comparison for LLM Fine-Tuning on SST-2 Dataset

In Table 5, we show the additional experiment results about **HiSo**'s acceleration on newer LLMs, including Qwen3-0.6B (Team, 2025b) and Gemma-3-270M (Team, 2025a). For DeComFL, we report the number of rounds required to fully converge. For **HiSo**, we report the rounds required to match DeComFL's best accuracy, together with the corresponding communication cost. Across all settings, **HiSo** consistently requires far fewer communication rounds while maintaining the same accuracy target. For the Qwen3-0.6B model, **HiSo** reduces the communication rounds from 2500 to 950 on SST-2 (a 2.6× speedup), from 1050 to 650 on QQP (a 1.6× speedup), and from 375 to 225 on SQuAD (a 1.7× speedup). This round reduction directly translates into lower communication cost. The acceleration is equally pronounced on the lighter Gemma-3-270M model. **HiSo** achieves a 2.1× speedup on SST-2, 1.4× on QQP, and 2× on SQuAD, again halving the communication cost compared to DeComFL. Overall, these results demonstrate that **HiSo** substantially accelerates zeroth-order FL, achieving 1.4×–2.6× faster convergence while preserving the extremely low communication footprint. This validates the key insight of **HiSo**: leveraging Hessian-informed ZO updates dramatically improves convergence efficiency while maintaining scalar-only communication.

Table 5: `HiSo`'s Acceleration. For DeComFL, we report the total number of communication rounds required to fully converge. For `HiSo`, we report the number of rounds needed to match DeComFL's best test accuracy, along with the corresponding communication cost.

| Model | Method | SST-2 | | | QQP | | | SQuAD | | |
|---|---|---|---|---|---|---|---|---|---|---|
| | | Round | Speedup | Comm. Cost | Round | Speedup | Comm. Cost | Round | Speedup | Comm. Cost |
| Qwen3-0.6B | DeComFL | 2500 | 1× | 83.16 KB | 1050 | 1× | 34.93 KB | 375 | 1× | 12.47 KB |
| | `HiSo` | 1000 | 2.5× | 33.26 KB | 650 | 1.6× | 21.62 KB | 225 | 1.7× | 7.48 KB |
| Gemma-3-270M | DeComFL | 2125 | 1× | 70.60 KB | 850 | 1× | 28.27 KB | 900 | 1× | 29.94 KB |
| | `HiSo` | 1025 | 2.1× | 34.05 KB | 625 | 1.4× | 20.79 KB | 450 | 2× | 14.97 KB |

## E.4 TIME COST RESULTS

### E.4.1 COMPUTATION TIME COST RESULTS

We profiled the actual computation time per round for Hessian-informed preconditioning ($T_{pre}$), along with the total computation time and estimated communication time on H100 GPUs. These results are presented in the following Table 6. Our profiling shows that the computation time for Hessian-informed preconditioning is negligible compared to both the overall computation and communication time. This confirms that `HiSo` remains scalable to LLMs. This efficiency is expected, as the core operations involved in our preconditioning step consist of a matrix squaring and matrix summation, both of which are computationally lightweight and do not scale prohibitively with model size.

In addition, the communication time heavily depends on the network condition and the number of transmitted scalars. To offer a comprehensive understanding, we provide an estimated communication time as follows: considering two types of common bandwidth: 100 Mbps (e.g., wifi or 5G) and 1 Gbps (e.g., enterprise LAN / wired campus network). If we run `HiSo` with 5 perturbations, it is reasonable to estimate that there are total 10 scalars to be transmitted in uplink and downlink per round. For transmitting 10 float32 scalars (approximately 1 KB including protocol overhead), the pure transmission time is about 0.08 ms under 100 Mbps, and 0.008 ms under 1 Gbps bandwidth. Including typical round-trip network latency, the total communication time per round is approximately 20-30 ms in 100 Mbps environments and 1-2 ms in 1 Gbps settings. This confirms that `HiSo`'s per-round preconditioning time (<0.5 ms in our experiments) is negligible compared to communication time.

Table 6: Comparison of Computation Time for Hessian-Informed Preconditioning with Different LLM Sizes (Using SST-2). $T_{pre}$ is the preconditioning time per round. $T_{total}$ is the total time per round. $T_{est}$ is the estimated communication time per round.

| Model Size | $T_{pre}$ | $T_{total}$ | $T_{pre}/T_{total}$ | $T_{est}$ |
|---|---|---|---|---|
| OPT-125M | 0.118 ms | 88.4 ms | 0.13% | 1∼30 ms |
| OPT-350M | 0.137 ms | 127.2 ms | 0.11% | 1∼30 ms |
| OPT-1.3B | 0.185 ms | 329.3 ms | 0.06% | 1∼30 ms |
| OPT-2.7B | 0.259 ms | 438.6 ms | 0.06% | 1∼30 ms |

### E.4.2 WALL-CLOCK TIME COST

In Table 7, we report the one-round computation time, one-round communication time and total wall-clock time until convergence of first-order baselines, ZO baseline and our `HiSo`. We observe that first-order FedAvg and FedAdam suffer from extremely high wall-clock time due to their high-dimensional communication overhead that each round requires 228.8 seconds, significantly dominating the overall runtime. Thus, despite their low per-round computation time, they require 68788 s and 45858 s total wall-clock time respectively.

In contrast, zeroth-order DeComFL significantly reduce communication to only $2.24 \times 10^{-6}$ seconds per round, achieving a 29× faster overall runtime than FedAdam. However, DeComFL still requires many more communication rounds to reach the target accuracy, compared to other three methods. Further, our HiSo achieves the best overall efficiency. Although the per-round computation time is slightly higher due to Hessian-informed updates, HiSo converges in far fewer rounds compared to DeComFL. Hence, HiSo achieves a total wall-clock time of only 1064 seconds, representing a 43× speedup over FedAdam, 64× speedup over FedAvg, and a 2.2× improvement over DeComFL. This shows that HiSo not only preserves the ultra-low communication cost of scalar-only communication

but also substantially accelerates convergence (i.e., less training rounds) through its Hessian-informed optimization design.

Table 7: Wall-Clock Time Cost Comparison (Qwen3-0.6B Model+SST-2 Dataset). We assume a typical 5G network setting with 100 Mbps uplink and 500 Mbps downlink bandwidth without considering latency, protocol and so on. We tested this on H100 GPUs.

| Method | One-round Computation Time | One-round Communication Time | Total Wall-Clock Time |
|---|---|---|---|
| FedAvg | 0.43 s | 228.8 s | 68788 s |
| FedAdam | 0.49 s | 228.8 s | 45858 s |
| DeComFL | 0.94 s | $2.24 \times 10^{-6}$ s | 2350 s |
| **HiSo** | 1.12 s | $2.24 \times 10^{-6}$ s | 1064 s |

### E.5 HiSo V.S. FL+PEFT BASELINES

Although this paper focuses on full-parameter FL, which differs in setup and assumptions from PEFT-based approaches, to provide a comprehensive evaluation, we include additional comparisons with FL+PEFT baselines. As shown in Table 8, **HiSo** achieves up to $10^4 \times$ communication overhead reduction while maintaining competitive test accuracy. Among ZO methods, **HiSo** consistently demonstrates lower communication overhead, higher test accuracy, and faster convergence.

Table 8: Comparison of **HiSo** and FL+PEFT Baselines. We report the communication cost of one client.

| Model | Dataset | Methods | Test Acc. | Comm. Cost |
|---|---|---|---|---|
| OPT-125M | SST-2 | FedAvg+LoRA (r=8) | 87.47% | 0.34 GB |
| | | FedSA-LoRA (Guo et al., 2025) (r=8) | 87.53% | 0.15 GB |
| | | FFA-LoRA (Sun et al., 2024) (r=8) | 87.39% | 0.16 GB |
| | | DeComFL+LoRA (Li et al., 2025b) (r=8) | 85.23% | 27.55 KB |
| | | **HiSo**+LoRA (Ours) (r=8) | 85.37% | 22.26 KB |
| | | **HiSo** (Ours) | 85.55% | 14.69 KB |

### E.6 TRAINING LOSS VS ITERATIONS/WALL-CLOCK TIME

Based on the convergence curves in Figure 8, **HiSo** consistently demonstrates substantially faster progress than DeComFL in both iterations and wall-clock time. On SST-2 with Qwen-3-0.6B, DeComFL requires roughly 2500 iterations to converge, while **HiSo** reaches the same accuracy within only about 1000 iterations, achieving a $2.5\times$ speedup in optimization rounds. This faster iteration-level convergence directly translates into end-to-end efficiency: when measuring real execution time, including communication and computation, **HiSo** is about $2.1\times$ faster than DeComFL in reaching the same final accuracy. The accuracy-time and loss-time curves further confirm that **HiSo** maintains a steeper descent trajectory throughout training, reducing both the number of updates and the total runtime needed for convergence. Overall, these results show that **HiSo** accelerates federated fine-tuning in both algorithmic efficiency (iterations) and practical efficiency (wall-clock time), providing significantly faster convergence under the same hardware and network conditions.

### E.7 SCENARIOS SUITABLE FOR HiSo

**HiSo** is not designed as a drop-in solution for any FL scenario. Its core characteristics, Zero-Order (ZO) nature and extreme communication efficiency, make it the best fit for these critical scenarios:

- **Gradient Inaccessibility.** When true gradients are inaccessible or prohibitively expensive to compute, zeroth-order (ZO) optimization serves as a natural and effective alternative. In such black-box settings, where only function evaluations are available, ZO methods enable optimization without explicit gradient information, thereby extending applicability to a broad range of tasks where gradient-based approaches are infeasible. As a ZO-FL method, **HiSo** is particularly well-suited to this scenario.

- **Bandwidth-Constrained Networks**. With a communication overhead limited to the kilobyte (KB) range, **HiSo** is ideally suited for FL deployments in environments with limited bandwidth. This feature guarantees scalability and practicality even when fine-tuning massive models.

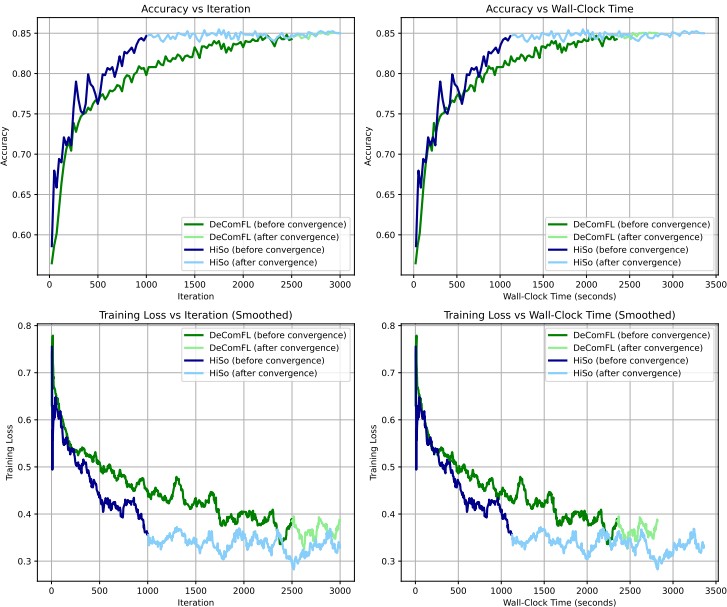

Figure 8: Training Loss vs Iteration and Training Loss vs Wall-Clock Time. (Qwen3-0.6B + SST-2)

# F  MAIN PROOF

## F.1  NOTATIONS

The following proof utilizes matrix and vector notations. A bold symbol, such as $\boldsymbol{x}_k$, generally represents a vector encompassing multiple clients, whereas a normal symbol, such as $x_k^{(i)}$, denotes the value for an individual client. To further lighten the notation for multiple clients and the local cost function, we adopt the following usage:

$$\boldsymbol{x}_k = \begin{bmatrix} x_k^{(1)} & x_k^{(2)} & \cdots & x_k^{(M)} \end{bmatrix} \in \mathbb{R}^{d \times M}, \tag{18}$$

$$\boldsymbol{f}(\boldsymbol{x}_k) = \begin{bmatrix} f_1(x_k^{(1)}; \xi_k^{(1)}) & f_2(x_k^{(2)}; \xi_k^{(1)}) & \cdots & f_M(x_k^{(M)}; \xi_k^{(1)}) \end{bmatrix} \in \mathbb{R}^{1 \times M}, \tag{19}$$

$$\nabla \boldsymbol{f}(\boldsymbol{x}_k) = \begin{bmatrix} \nabla f_1(x_k^{(1)}; \xi_k^{(1)}) & \nabla f_2(x_k^{(2)}; \xi_k^{(1)}) & \cdots & \nabla f_M(x_k^{(M)}; \xi_k^{(1)}) \end{bmatrix} \in \mathbb{R}^{d \times M}. \tag{20}$$

where $\nabla f_1(x_k^{(1)}; \xi_k^{(1)})$ represent the stochastic gradient evaluated on local cost function $f_1$ at the point $x_k^{(1)}$. Notice the function value $f_i$ or the gradient $\nabla f_i$ applied on the different iterates $\boldsymbol{x}_k^{(i)}$ in above notations. Various vector and matrix norms are used in the proof. For any semi-positive definite matrix $\Sigma$, we adopt the following convention in Table 9.

Table 9: Norm Notations in This Paper

| Notation | Definition | Comment |
|---|---|---|
| $\|x\|_\Sigma^2$ | $x^\mathsf{T} \Sigma x$ | Mahalanobis (weighted) vector norm, where $x \in \mathbb{R}^d$. |
| $\|A\|_\Sigma^2$ | $\mathrm{Tr}(A^\mathsf{T} \Sigma A)$ | Mahalanobis (weighted) matrix norm $A \in \mathbb{R}^{d \times d}$ |
| $\|A\|_2, \|A\|$ | $\sigma_{\max}(A)$ | Spectrum norm, i.e., largest singular value of $A$ |
| $\|\boldsymbol{x}\|_F^2$ | $\mathrm{Tr}(\boldsymbol{x}^\mathsf{T} \boldsymbol{x})$ | Frobenius norm (note $\boldsymbol{x}$ is matrix here) |

**Remark:** While the Frobenius norm can be viewed as a special case of the weighted matrix norm, confusion is unlikely in this paper as we only apply the Frobenius norm to the stacked vector $\boldsymbol{x}$.

Other commonly used constants and symbols are summarized in the following table.

Table 10: Notations in This Paper

| Notation | Meaning |
| --- | --- |
| $i$ | Index of clients |
| $k$ | Index of iterations |
| $r$ | Index of communication round and $r = \lfloor k/\tau \rfloor \tau$ |
| $\tau$ | The number of local update steps |
| $C_r$ | Indices set of clients sampled at $r$-th round |
| $d$ | Model parameter dimension |
| $m, M$ | Number of sampled and total clients |
| $f_i, F$ | Local and global loss function |
| $u, z$ | A random vector drawing from the standard and weighted Gaussian distributions |

The all-one vector $\mathbb{1} = [1, 1, \cdots, 1]^\mathsf{T} \in \mathbb{R}^{M \times 1}$ and the uniform vector $\mathbb{1}_u = \mathbb{1}/M \in \mathbb{R}^{M \times 1}$ are two common notations we adopted in the rest of the proof. With these symbols, we have the following identity

$$\nabla \boldsymbol{f}(x\mathbb{1}^\mathsf{T})\mathbb{1}_u = \nabla F(x) \in \mathbb{R}^{d \times 1} \tag{21}$$

## F.2 ALGORITHM REFORMULATION AND MAIN RECURSION

To make a concise proof, we first re-write the algorithm into the vector-matrix form as introduced in the previous section. First, to make the convergence proof straightforward, we translate the two-level for-loop structure (outer round loop and inner local update loop) into a single recursion structure. The $k$-th local update in $r$-th communication round is equivalent to the $r\tau + k$ iterations. Then, inspired by the work (Li et al., 2020; Ying et al., 2025), first we notice the Federated Learning algorithm is equivalent if we virtually send the server's model to all clients but keep the aggregation step the same, i.e., only aggregate the clients' values in $C_r$. Under this form, we can equivalently reformulate the algorithm into this recursion

$$\boldsymbol{y}_{k+1} = \boldsymbol{x}_k - \eta H_k^{-1/2} u_k \frac{\boldsymbol{f}(\boldsymbol{x}_k + \mu H_k^{-1/2} u_k \mathbb{1}^\mathsf{T}) - \boldsymbol{f}(\boldsymbol{x}_k)}{\mu}, \tag{22}$$

$$\boldsymbol{x}_{k+1} = \boldsymbol{y}_{k+1} W_k. \tag{23}$$

where $\boldsymbol{x}_k, \boldsymbol{y}_k \in \mathbb{R}^{d \times M}$ is the stacked vectors and $W_k$ represents the communication matrix. Note the single subscript $k$ is for the iteration, which is not the same $k$ in the double subscripts for local update step. The element of $W_k[i, j]$ represents the effective weight that client $i$ to client $j$ at iteration $k$. If the iteration $k \neq r\tau$, $W_k = I$ – local update step. If $k = r\tau$, $W_k$ becomes some average matrix representing the model average step. More concretely, it is a **column-stochastic** matrix, each column having the same weights and the non-zero elements in each column are the sampled clients in round $r$. For instance, suppose client $\{0, 1, 3\}$ sampled in the four clients case, the corresponding $W_k$ are

$$W_k = \begin{bmatrix} \frac{1}{3} & \frac{1}{3} & \frac{1}{3} & \frac{1}{3} \\ 0 & 0 & 0 & 0 \\ \frac{1}{3} & \frac{1}{3} & \frac{1}{3} & \frac{1}{3} \\ \frac{1}{3} & \frac{1}{3} & \frac{1}{3} & \frac{1}{3} \end{bmatrix} \tag{24}$$

Back to the update rule (22) – (23), the following proof is for the general update rule of $H_k$. Hence, we just need to focus on the property of $H_k$ instead of combining the update rule and revisit it later. We further denote $z_k = H_k^{-1/2} u_k, z_k \sim \mathcal{N}(0, H_k^{-1})$ to simplify the update rule:

$$\boldsymbol{y}_{k+1} = \boldsymbol{x}_k - \frac{\eta}{\mu} z_k \Big( \boldsymbol{f}(\boldsymbol{x}_k + \mu z_k \mathbb{1}^\mathsf{T}) - \boldsymbol{f}(x_k) \Big), \tag{25}$$

$$\boldsymbol{x}_{k+1} = \boldsymbol{y}_{k+1} W_k \tag{26}$$

Because of the shared seeds and Hessians, $\boldsymbol{z}_k$ is a variable that has no client index subscripts. Using directional gradient approximation

$$f(x + \mu z) = f(x) + \mu z^{\mathsf{T}} \nabla f(x) + \frac{\mu^2}{2} z^{\mathsf{T}} \left( \int_0^1 \nabla^2 f(x + tz) dt \right) z, \tag{27}$$

the update rule can be concisely written as

$$\boldsymbol{y}_{k+1} = \boldsymbol{x}_k - \eta z_k z_k^{\mathsf{T}} \nabla \boldsymbol{f}(\boldsymbol{x}_k) + O(\mu\eta), \tag{28}$$

$$\boldsymbol{x}_{k+1} = \boldsymbol{y}_{k+1} W_k, \tag{29}$$

**To manage notational complexity and the handling of intricate coefficients, we adopt the** $O(\mu\eta)$ **notation.** Since this paper concentrates on addressing client sampling and local updates in federated learning, the analysis of the zeroth-order approximation error is intentionally simplified. This approach facilitates a clearer understanding of the distinct error sources in the federated setting, without sacrificing proof rigor.

We define the (virtual) centralized iterates $\bar{x}_k := \boldsymbol{x}_k \mathbb{1}_u$ and $\bar{y}_k := \boldsymbol{y}_k \mathbb{1}_u$. The recursion of centralized iterates $\bar{x}_k := \boldsymbol{x}_k \mathbb{1}_u$ is

$$\bar{x}_{k+1} = \boldsymbol{y}_{k+1} W_k \mathbb{1}_u \tag{30}$$

$$= \left( \boldsymbol{x}_k - \eta z_k z_k^{\mathsf{T}} \nabla \boldsymbol{f}(\boldsymbol{x}_k) \right) w_k + O(\mu\eta) \tag{31}$$

where we define $w_k := W_k \mathbb{1}_u$. It is straightforward to see that if $k \neq r\tau$, $w_k = \mathbb{1}_u$; if $k = r\tau$, $w_k$ is the random selection vector with each entry having $m/M$ probability to be $1/m$ and 0 otherwise. Hence, we have the following two cases to handle with

$$\bar{x}_{k+1} = \begin{cases} \bar{x}_k - \eta z_k z_k^{\mathsf{T}} \overline{\nabla \boldsymbol{f}}(\boldsymbol{x}_k) + O(\mu\eta) & k \neq r\tau, \\ \hat{x}_k - \eta z_k z_k^{\mathsf{T}} \widehat{\nabla \boldsymbol{f}}(\boldsymbol{x}_k) + O(\mu\eta) & k = r\tau. \end{cases} \tag{32}$$

where we denote

$$\hat{x}_k = \boldsymbol{x}_k w_k, \tag{33}$$

$$\overline{\nabla \boldsymbol{f}}(\boldsymbol{x}_k) = \nabla \boldsymbol{f}(\boldsymbol{x}_k) \mathbb{1}_u = \frac{1}{M} \sum_{i=1}^{M} \nabla f_i(x_k^{(i)}) \in \mathbb{R}^{d \times 1}, \tag{34}$$

$$\widehat{\nabla \boldsymbol{f}}(\boldsymbol{x}_k) = \nabla \boldsymbol{f}(\boldsymbol{x}_k) w_k = \frac{1}{m} \sum_{i \in C_r} \nabla f_i(x_k^{(i)}) \in \mathbb{R}^{d \times 1}. \tag{35}$$

Above two centralized recursions will be the main reference the following proof.

## F.3 KEY LEMMAS

### F.3.1 LEMMAS ABOUT GAUSSIAN VARIABLES

The rest proof is built on top of the following two fundamental lemmas about the Gaussian distribution.

**Lemma 1** (Fourth-Order Moment of Gaussian Vector). *Suppose that the random vector $z \sim \mathcal{N}(0, \Lambda)$ where $\Lambda$ is a diagonal matrix. For any symmetric matrix $W$, we have*

$$\mathbb{E} zz^{\mathsf{T}} W zz^{\mathsf{T}} = \mathrm{Tr}(W\Lambda) \cdot \Lambda + 2\Lambda W \Lambda. \tag{36}$$

*If $u \sim \mathcal{N}(0, I)$, i.e., drawing from a standard Gaussian distribution, we have*

$$\mathbb{E} uu^{\mathsf{T}} W uu^{\mathsf{T}} = \mathrm{Tr}(W) \cdot I + 2W. \tag{37}$$

*Proof.* Let the matrix $\Psi = zz^{\mathsf{T}} W zz^{\mathsf{T}}$. For each element $i \neq j$,

$$\Psi[i,j] = \mathbb{E} z_i z_j (\sum_{i',j'} z_{i'} z_{j'} W[i',j']) = 2\mathbb{E} z_i^2 z_j^2 W[i,j] = 2\Lambda_i \Lambda_j W[i,j], \tag{38}$$

where the second equality holds because the zero-mean property of $z$ and $z_i$ is independent of each other. For the diagonal elements,

$$\Psi[i,i] = \mathbb{E} z_i^2 (\sum_{i',j;} z_{i'} z_{j'} W[i',j']) = \sum_{i'} \mathbb{E} z_i^2 z_{i'}^2 W[i',i']$$

$$= \sum_{i' \neq i} \mathbb{E}\, z_i^2 \mathbb{E}\, z_{i'}^2 W[i', i'] + \mathbb{E}\, z_i^4 W[i, i]$$

$$= \Lambda_i \sum_{i'} \Lambda_{i'} W[i', i'] + 2W[i, i]\Lambda_i^2, \tag{39}$$

where we utilize the fact that $\mathbb{E}\, z_i^4 = 3\Lambda_i^2$. Lastly, combining the above two results into a concise matrix notation, we establish

$$\Psi = \text{Tr}(W\Lambda) \cdot \Lambda + 2\Lambda W \Lambda \tag{40}$$

For the standard Gaussian distribution case, we just need to substitute $\Lambda = I$ into equation 36. $\quad\square$

**Lemma 2** (Gaussian Smoothed Function). *We define a smooth approximation of objective function $f$ as $f^\mu(\cdot)$ that can be formulated as*

$$f^\mu(x) := \frac{1}{(2\pi)^{\frac{d}{2}}} \int f(x + \mu u) e^{-\frac{1}{2}\|u\|^2} d\mathbf{z} = \mathbb{E}\left[f(x + \mu)\right], \tag{41}$$

*where $\mu > 0$ is the smoothing parameter, and $\mathbf{z}$ is one n-dimensional standard Gaussian random vector. Then, we have*

$$\mathbb{E}\, \frac{f(x + \mu u) - f(x)}{\mu} u = \nabla f^\mu(x), \quad \text{where } u \sim \mathcal{N}(0, I) \tag{42}$$

*Above equality implies the ZO gradient is an unbiased estimate of the gradient of the smoothed function $f^\mu$.*

*Proof.* See the proof in (Ghadimi & Lan, 2013; Nesterov & Spokoiny, 2017). $\quad\square$

### F.3.2 VARIANCE LEMMA FOR SAMPLING NOISE

Before we present the main proof, we first bound the variance of $\widehat{\nabla f}(\mathbf{x}_k)$.

**Lemma 3.** *Suppose $f_i$ is L-smooth and the local cost functions satisfy the data heterogeneity assumption $\sigma_G^2$. For any semi-positive definite matrix $\Sigma$, the variance of the sampled gradient $\widehat{\nabla f}(\mathbf{x}_k)$ satisfies:*

$$\mathbb{E}\|\widehat{\nabla f}(\mathbf{x}_k)\|_\Sigma^2 \leq 2\|\nabla F(\bar{x}_k)\|_\Sigma^2 + \frac{2}{m}\|\Sigma\|(\sigma_G^2 + \sigma_s^2) + \frac{2L^2}{M}\|\Sigma\|\|\mathbf{x}_k - \bar{x}_k \mathbf{1}^\top\|_F^2, \tag{43}$$

*where $m$ is the number of sampled clients per round and $M$ is the total number of clients.*

*Proof.* For any semi-positive matrix $\Sigma$, we have

$$\mathbb{E}\|\widehat{\nabla f}(\mathbf{x}_k)\|_\Sigma^2 \leq 2\mathbb{E}\|\widehat{\nabla f}(\bar{x}_k \mathbf{1}^\top)\|_\Sigma^2 + 2\mathbb{E}\|\widehat{\nabla f}(\mathbf{x}_k) - \widehat{\nabla f}(\bar{x}_k \mathbf{1}^\top)\|_\Sigma^2 \tag{44}$$

where the inequality utilizes Jensen's inequality.

Next, noticing that the variance identity for any weighted distance $\|\cdot\|_\Sigma$ satisfies

$$\mathbb{E}\|\bar{x}_k - \mathbb{E}\,\bar{x}_k\|_\Sigma^2 = \mathbb{E}\|\bar{x}_k\|_\Sigma^2 - \mathbb{E}(\bar{x}_k^\top \Sigma \mathbb{E}\,\bar{x}_k) - \mathbb{E}(\mathbb{E}\,\bar{x}_k^\top)\Sigma\bar{x}_k + \|\mathbb{E}\,\bar{x}_k\|_\Sigma^2$$

$$= \mathbb{E}\|\bar{x}_k\|_\Sigma^2 - \|\mathbb{E}\,\bar{x}_k\|_\Sigma^2 \tag{45}$$

Combining with the fact that $\mathbb{E}_{w_k} \widehat{\nabla f}(\bar{x}_k \mathbf{1}^\top) = \nabla F(\bar{x}_k)$, we establish

$$\mathbb{E}\|\widehat{\nabla f}(\bar{x}_k \mathbf{1}^\top)\|_\Sigma^2 = \mathbb{E}\|\widehat{\nabla f}(\bar{x}_k \mathbf{1}^\top) - \nabla F(\bar{x}_k)\|_\Sigma^2 + \|\nabla F(\bar{x}_k)\|_\Sigma^2 \tag{46}$$

The first term in the above equality can be further bounded through the data heterogeneity assumption that

$$\mathbb{E}\|\widehat{\nabla f}(\bar{x}_k \mathbf{1}^\top) - \nabla F(\bar{x}_k)\|_\Sigma^2 = \frac{1}{m^2}\mathbb{E}\Big\|\sum_{i \in C_r}\Big(\nabla f_i(\bar{x}_k; \xi_k) - \nabla F(\bar{x}_k)\Big)\Big\|_\Sigma^2$$

$$= \frac{1}{mM}\sum_{i=1}^{M}\|\nabla f_i(\bar{x}_k; \xi_k) - \nabla F(\bar{x}_k)\|_\Sigma^2$$

$$\leq \frac{1}{m}\|\Sigma\|(\sigma_G^2 + \sigma_s^2) \tag{47}$$

where the second equality holds since the zero-mean property. Substituting the above results back to equation 44, we arrive

$$\mathbb{E}\|\widehat{\nabla f}(\boldsymbol{x}_k)\|_\Sigma^2 \leq 2\|\nabla F(\bar{x}_k)\|_\Sigma^2 + \frac{2}{m}\|\Sigma\|(\sigma_G^2 + \sigma_s^2) + 2\mathbb{E}\|\widehat{\nabla f}(\boldsymbol{x}_k) - \widehat{\nabla f}(\bar{x}_k\mathbb{1}^\mathsf{T})\|_\Sigma^2$$

$$\leq 2\|\nabla F(\bar{x}_k)\|_\Sigma^2 + \frac{2}{m}\|\Sigma\|(\sigma_G^2 + \sigma_s^2) + 2L^2\|\Sigma\|\|\boldsymbol{x}_k - \bar{x}_k\mathbb{1}^\mathsf{T}\|_F^2/M \qquad (48)$$

where we applied the $L-$ Lipschitz condition and Jensen's inequality in the last step. $\qquad\square$

## F.4 DESCENT LEMMA

**Lemma 4.** *When* $\eta \leq \left\{\frac{\beta_\ell}{mL}, \frac{1}{8\rho_k}\right\}$, *the virtual centralized iterates* $\bar{x}_k$ *of one round satisfy*

$$\mathbb{E}\,F(\bar{x}_{(r+1)\tau+1}) \leq \mathbb{E}\,F(\bar{x}_{r\tau+1}) - \frac{\eta}{4}\sum_{j=r\tau+1}^{(r+1)\tau}\|\nabla F(\bar{x}_j)\|_{H_r^{-1}}^2 + O(\eta^2\mu)$$

$$+ \frac{4\tau\eta^2}{\beta_\ell m}\sum_{j=r\tau+1}^{(r+1)\tau}\rho_k(\sigma_G^2 + \sigma_s^2) + \frac{2L}{mM}\sum_{j=r\tau+1}^{(r+1)\tau}\|\boldsymbol{x}_j - \bar{x}_j\mathbb{1}^\mathsf{T}\|_F^2 \qquad (49)$$

*where* $\rho_k = \mathrm{Tr}(H_k^{-1/2}\Sigma_k H_k^{-1/2}) + 2\|H_k^{-1/2}\Sigma_k H_k^{-1/2}\|$. $\qquad\square$

**Proof.** Recall there are two random variables in the main recursion Eq. (32), one is the ZO random direction $z_k$ and the other is the client sampling vector $w_k$. First, taking the conditional expectation over $w_k$, we have

$$\mathbb{E}_{w_k}\bar{x}_{k+1} = \bar{x}_k - \eta z_k z_k^\mathsf{T}\overline{\nabla f}(\boldsymbol{x}_k) + \mathcal{O}(\eta\mu) \qquad (50)$$

for any iteration $k$. Then, taking conditional expectation over $z_k$, we have

$$\mathbb{E}\,\bar{x}_{k+1} = \bar{x}_k - \eta H_k^{-1}\overline{\nabla f}(\boldsymbol{x}_k) + \mathcal{O}(\eta\mu) \qquad (51)$$

As a result of Assumption 1, there is a semi-positive definite matrix $\Sigma_y \preceq L \cdot I_d$ such that the global loss function satisfies

$$F(x) \leq F(y) + \langle \nabla F(y), x - y\rangle + \frac{1}{2}(x - y)^\mathsf{T}\Sigma_y(x - y). \qquad (52)$$

Hence, we have

$$F(\bar{x}_{k+1}) \leq F(\bar{x}_k) + \langle \nabla F(\bar{x}_k), \bar{x}_{k+1} - \bar{x}_k\rangle + \frac{1}{2}(\bar{x}_{k+1} - \bar{x}_k)^\mathsf{T}\Sigma_k(\bar{x}_{k+1} - \bar{x}_k) \qquad (53)$$

Now, substituting Eq. (32) into the above expansion and taking the conditional expectation, we will establish the following two cases.

**Local Update Iteration:**

When the iteration $k$ is not the communication iteration, i.e. $k \neq r\tau$, we have

$$\mathbb{E}\,F(\bar{x}_{k+1}) \leq F(\bar{x}_k) - \eta\overline{\nabla f}(\boldsymbol{x}_k)^\mathsf{T}H_k^{-1}\nabla F(\bar{x}_k) + O(\eta^2\mu)$$

$$+ \eta^2\mathbb{E}[\widehat{\nabla f}(\boldsymbol{x}_k)^\mathsf{T}z_k z_k^\mathsf{T}\Sigma_k z_k z_k^\mathsf{T}\widehat{\nabla f}(\boldsymbol{x}_k)] \qquad (54)$$

First, we focus on the cross term

$$-\overline{\nabla f}(\boldsymbol{x}_k)^\mathsf{T}H_k^{-1}\nabla F(\bar{x}_k) = -\nabla F(\bar{x}_k)^\mathsf{T}H_k^{-1}\nabla F(\bar{x}_k) + (\nabla F(\bar{x}_k) - \overline{\nabla f}(\boldsymbol{x}_k))^\mathsf{T}H_k^{-1}\nabla F(\bar{x}_k)$$

$$\leq -\|\nabla F(\bar{x}_k)\|_{H_k^{-1}}^2 + \frac{1}{2}\|\nabla F(\bar{x}_k)\|_{H_k^{-1}}^2 + \frac{1}{2}\|\nabla F(\bar{x}_k) - \overline{\nabla f}(\boldsymbol{x}_k)\|_{H_k^{-1}}^2$$

$$= -\frac{1}{2}\|\nabla F(\bar{x}_k)\|_{H_k^{-1}}^2 + \frac{1}{2}\|\nabla F(\bar{x}_k) - \overline{\nabla f}(\boldsymbol{x}_k)\|_{H_k^{-1}}^2 \qquad (55)$$

Because of Assumption 4, we have $\beta_u^{-1} \leq \|H_k^{-1}\| \leq \beta_\ell^{-1}$, which implies

$$\frac{1}{2}\|\nabla F(\bar{x}_k) - \overline{\nabla f}(\boldsymbol{x}_k)\|_{H_k^{-1}}^2 \leq \frac{1}{2\beta_\ell}\|\nabla F(\bar{x}_k) - \overline{\nabla f}(\boldsymbol{x}_k)\|^2$$

$$\leq \frac{1}{2\beta_\ell N}\sum_{i=1}^{M}\|\nabla f_i(\bar{x}_k) - \nabla f_i(x_k^{(i)})\|^2$$

$$= \frac{L^2}{2\beta_\ell N}\|\boldsymbol{x}_k - \bar{x}_k \mathbb{1}^\mathsf{T}\|_F^2 \tag{56}$$

Substituting back, we have

$$\mathbb{E}\, F(\bar{x}_{k+1}) \leq F(\bar{x}_k) - \frac{\eta}{2}\|\nabla F(\bar{x}_k)\|_{H_k^{-1}}^2 + \frac{\eta L^2}{2\beta_\ell N}\|\boldsymbol{x}_k - \bar{x}_k \mathbb{1}^\mathsf{T}\|_F^2$$
$$+ \eta^2 \underbrace{\mathbb{E}\,[\widehat{\nabla \boldsymbol{f}}(\boldsymbol{x}_k)^\mathsf{T} z_k z_k^\mathsf{T} \Sigma_k z_k z_k^\mathsf{T} \widehat{\nabla \boldsymbol{f}}(\boldsymbol{x}_k)]}_{:=Q} \tag{57}$$

Next, the key is this quadratic term. Leveraging Lemma 1, we establish

$$Q = \mathbb{E}_{w_k}\left(\widehat{\nabla \boldsymbol{f}}(\boldsymbol{x}_k)^\mathsf{T}\left(\mathrm{Tr}(\Sigma_k H_k^{-1})H_k^{-1} + 2H_k^{-1}\Sigma_k H_k^{-1}\right)\widehat{\nabla \boldsymbol{f}}(\boldsymbol{x}_k)\right)$$
$$\leq (\mathrm{Tr}(\Sigma_k H_k^{-1}) + 2\|H^{-1/2}\Sigma_k H^{-1/2}\|)\mathbb{E}_{w_k}\|\widehat{\nabla \boldsymbol{f}}(\boldsymbol{x}_k)\|_{H_k^{-1}}^2 \tag{58}$$

where we utilize the following inequality in the last step

$$\|x\|_{H_k^{-1}\Sigma_k H_k^{-1}}^2 = \mathrm{Tr}(H_k^{-1/2}xx^\mathsf{T} H_k^{-1/2}H_k^{-1/2}\Sigma_k H_k^{-1/2}) \leq \|H_k^{-1/2}\Sigma_k H_k^{-1/2}\|\|x\|_{H_k^{-1}}^2.$$

For simplicity, we introduce the matrix $\Xi_k = H_k^{-1/2}\Sigma_k H_k^{-1/2}$. Plugging the previous sampling noise variance result (48), we establish

$$Q \leq (\mathrm{Tr}(\Xi_k) + 2\|\Xi_k\|)\left(2\|\nabla F(\bar{x}_k)\|_{H_k^{-1}}^2 + \frac{2}{\beta_\ell m}(\sigma_G^2 + \sigma_s^2) + \frac{2L^2}{\beta_\ell M}\|\boldsymbol{x}_k - \bar{x}_k \mathbb{1}^\mathsf{T}\|_F^2/M\right) \tag{59}$$

This $\mathrm{Tr}(\Xi_k) + 2\|\Xi_k\|$ is the key quantity that we will encounter repeatedly. To further reduce the notation, we denote $\rho_k = \mathrm{Tr}(\Xi_k) + 2\|\Xi_k\|$ Combining all the above results, we have

$$\mathbb{E}\, F(\bar{x}_{k+1}) \leq F(\bar{x}_k) - \left(\frac{\eta}{2} - 2\eta^2\rho_k\right)\|\nabla F(\bar{x}_k)\|_{H_k^{-1}}^2 + O(\eta^2\mu)$$
$$+ \left(\frac{\eta L^2}{2\beta_\ell M} + \frac{2\eta^2 L^2\rho_k}{\beta_\ell M}\right)\|\boldsymbol{x}_k - \bar{x}_k \mathbb{1}^\mathsf{T}\|_F^2 + \frac{2\eta^2\rho_k}{\beta_\ell m}(\sigma_G^2 + \sigma_s^2) \tag{60}$$

When $\eta \leq \frac{1}{4\rho_k}$, the coefficients can be simplified into

$$\mathbb{E}\, F(\bar{x}_{k+1}) \leq F(\bar{x}_k) - \frac{\eta}{4}\|\nabla F(\bar{x}_k)\|_{H_k^{-1}}^2 + O(\eta^2\mu)$$
$$+ \frac{\eta L^2}{\beta_\ell M}\|\boldsymbol{x}_k - \bar{x}_k \mathbb{1}^\mathsf{T}\|_F^2 + \frac{2\eta^2\rho_k}{\beta_\ell m}(\sigma_G^2 + \sigma_s^2) \tag{61}$$

**Communication Iteration:**

When the iteration $k$ is the communication iteration, i.e. $k \neq r\tau$, we have

$$\mathbb{E}\, F(\bar{x}_{k+1}) \leq F(\bar{x}_k) - \eta\overline{\nabla \boldsymbol{f}}(\boldsymbol{x}_k)^\mathsf{T} H_k^{-1}\nabla F(\bar{x}_k) + O(\eta^2\mu)$$
$$+ \mathbb{E}\left(\hat{x}_k - \bar{x}_k - \eta\eta z_k z_k^\mathsf{T}\widehat{\nabla \boldsymbol{f}}(\boldsymbol{x}_k)\right)^\mathsf{T}\Sigma_k\left(\hat{x}_k - \bar{x}_k - \eta\eta z_k z_k^\mathsf{T}\widehat{\nabla \boldsymbol{f}}(\boldsymbol{x}_k)\right)$$
$$\leq F(\bar{x}_k) - \eta\overline{\nabla \boldsymbol{f}}(\boldsymbol{x}_k)^\mathsf{T} H_k^{-1}\nabla F(\bar{x}_k) + O(\eta^2\mu)$$
$$+ 2\mathbb{E}\,(\hat{x}_k - \bar{x}_k)^\mathsf{T}\Sigma_k(\hat{x}_k - \bar{x}_k) + 2\eta^2\mathbb{E}\,[\widehat{\nabla \boldsymbol{f}}(\boldsymbol{x}_k)^\mathsf{T} z_k z_k^\mathsf{T}\Sigma_k z_k z_k^\mathsf{T}\widehat{\nabla \boldsymbol{f}}(\boldsymbol{x}_k)] \tag{62}$$

Next, we notice that

$$\mathbb{E}\,(\hat{x}_k - \bar{x}_k)^\mathsf{T}\Sigma_k(\hat{x}_k - \bar{x}_k) \leq L\mathbb{E}\,\|\hat{x}_k - \bar{x}_k\|^2 = \frac{L}{mM}\|\boldsymbol{x}_k - \bar{x}_k \mathbb{1}^\mathsf{T}\|_F^2 \tag{63}$$

Utilizing previously established result Eq. (60), we have

$$\mathbb{E}\, F(\bar{x}_{k+1}) \leq F(\bar{x}_k) - \left(\frac{\eta}{2} - 4\eta^2\rho_k\right)\|\nabla F(\bar{x}_k)\|_{H_k^{-1}}^2 + O(\eta^2\mu)$$
$$+ \left(\frac{L}{m} + \frac{\eta L^2}{2\beta_\ell} + \frac{4\eta^2 L^2}{\beta_\ell M}\rho_k\right)\|\boldsymbol{x}_k - \bar{x}_k \mathbb{1}^\mathsf{T}\|_F^2 + \frac{4\eta^2\rho_k}{\beta_\ell m}(\sigma_G^2 + \sigma_s^2) \tag{64}$$

When $\eta \leq \frac{1}{8\rho_k}$, the coefficients can be simplified into

$$\mathbb{E}\, F(\bar{x}_{k+1}) \leq F(\bar{x}_k) - \frac{\eta}{4}\|\nabla F(\bar{x}_k)\|_{H_k^{-1}}^2 + O(\eta^2\mu)$$
$$+ \left(\frac{L}{mM} + \frac{\eta L^2}{\beta_\ell M}\right)\|\boldsymbol{x}_k - \bar{x}_k \mathbb{1}^\mathsf{T}\|_F^2 + \frac{4\eta^2\rho_k}{\beta_u m}(\sigma_G^2 + \sigma_s^2) \tag{65}$$

We further require the learning rate $\eta \leq \frac{\beta_\ell}{mL}$ to establish

$$
\begin{aligned}
\mathbb{E}\, F(\bar{x}_{k+1}) \leq & F(\bar{x}_k) - \frac{\eta}{4}\|\nabla F(\bar{x}_k)\|^2_{H_k^{-1}} + O(\eta^2\mu) \\
& + \frac{2L}{mM}\|\boldsymbol{x}_k - \bar{x}_k\mathbb{1}^\mathsf{T}\|^2_F + \frac{4\eta^2\rho_k}{\beta_\ell m}(\sigma_G^2 + \sigma_s^2)
\end{aligned}
\tag{66}
$$

**Combining Two into One Round:**

Combining the above two results and iterating from $k = r\tau + 1$ to $k = (r+1)\tau$, we establish

$$
\begin{aligned}
\mathbb{E}\, F(\bar{x}_{(r+1)\tau+1}) \leq & \mathbb{E}\, F(\bar{x}_{r\tau+1}) - \frac{\eta}{4}\sum_{j=r\tau+1}^{(r+1)\tau}\|\nabla F(\bar{x}_j)\|^2_{H_r^{-1}} + O(\eta^2\mu) \\
& + \frac{4\tau\eta^2\rho_k}{\beta_\ell m}(\sigma_G^2 + \sigma_s^2) + \frac{2L}{mM}\sum_{j=r\tau+1}^{(r+1)\tau}\|\boldsymbol{x}_j - \bar{x}_j\mathbb{1}^\mathsf{T}\|^2_F,
\end{aligned}
\tag{67}
$$

where we can absorb the coefficients on the consensus term $\|\boldsymbol{x}_j - \bar{x}_j\mathbb{1}^\mathsf{T}\|^2_F$ into $2L/mM$ since above we already require the learning rate $\eta \leq \frac{\beta_\ell}{mL}$. Also, we replace $H_k$ by $H_r$ since it is not updated within one communication round. $\qquad\square$

### F.5   CONSENSUS LEMMA

**Lemma 5.** *When $\eta \leq \frac{\beta_\ell}{4(\tau-1)}\sqrt{\frac{1}{L(d+2)}}$, the sum of the consensus error of one round is bounded by the following term*

$$
\frac{1}{\tau}\sum_{k=r\tau+1}^{(r+1)\tau}\mathbb{E}\,\|\boldsymbol{x}_k - \bar{x}_k\mathbb{1}^\mathsf{T}\|^2_F \leq 4\eta^2(\tau-1)^2 M\beta_\ell^{-1}\|\Phi_r\|(\sigma_G^2 + \sigma_s^2) + O(\eta^2\mu^2)
\tag{68}
$$

*where $\Phi_r := \mathrm{Tr}(H_r^{-1}) + 2H_r^{-1}$.* $\qquad\square$

**Proof.** The consensus residual is defined as

$$
\|\boldsymbol{x}_{k+1} - \bar{x}_{k+1}\mathbb{1}^\mathsf{T}\|^2_F = \|\boldsymbol{x}_k - \bar{x}_k\mathbb{1}^\mathsf{T} - \eta(z_k z_k^\mathsf{T}\nabla\boldsymbol{f}(\boldsymbol{x}_k) - z_k z_k^\mathsf{T}\nabla\boldsymbol{f}(\boldsymbol{x}_k)\mathbb{1}_u\mathbb{1}^\mathsf{T}) + O(\eta\mu)\|^2_F
\tag{69}
$$

If $k = r\tau$, all clients have the same value. Hence, we can expand the difference $\boldsymbol{x}_k - \bar{x}_k\mathbb{1}^\mathsf{T}$ up to $k = r\tau$ and arrive at

$$
\begin{aligned}
& \|\boldsymbol{x}_{k+1} - \bar{x}_{k+1}\mathbb{1}^\mathsf{T}\|^2_F \\
& = \left\|\eta\sum_{j=r\tau+1}^{k}\left(z_j z_j^\mathsf{T}\nabla\boldsymbol{f}(\boldsymbol{x}_j) - z_j z_j^\mathsf{T}\nabla\boldsymbol{f}(\boldsymbol{x}_j)\mathbb{1}_u\mathbb{1}^\mathsf{T}\right) + O(\eta\mu)\right\|^2_F \\
& \leq (\tau-1)\sum_{j=r\tau+1}^{k}\eta^2\|z_j z_j^\mathsf{T}\nabla\boldsymbol{f}(\boldsymbol{x}_j) - z_j z_j^\mathsf{T}\nabla\boldsymbol{f}(\boldsymbol{x}_j)\mathbb{1}_u\mathbb{1}^\mathsf{T}\|^2_F + O(\eta^2\mu^2),
\end{aligned}
\tag{70}
$$

where we utilize Jensen's inequality in the above step. Next, we focus on the term in the summation

$$
\begin{aligned}
& \|z_j z_j^\mathsf{T}\nabla\boldsymbol{f}(\boldsymbol{x}_j) - z_j z_j^\mathsf{T}\nabla\boldsymbol{f}(\boldsymbol{x}_j)\mathbb{1}_u\mathbb{1}^\mathsf{T}\|^2_F \\
& \leq 4\|z_j z_j^\mathsf{T}\nabla\boldsymbol{f}(\boldsymbol{x}_j) - z_j z_j^\mathsf{T}\nabla\boldsymbol{f}(\bar{x}_j\mathbb{1}^\mathsf{T})\|^2_F + 2\|z_j z_j^\mathsf{T}\nabla\boldsymbol{f}(\bar{x}_j\mathbb{1}^\mathsf{T}) - z_j z_j^\mathsf{T}\nabla F(\bar{x}_j\mathbb{1}^\mathsf{T})\mathbb{1}^\mathsf{T}\|^2_F \\
& \quad + 4\|z_j z_j^\mathsf{T}\nabla\boldsymbol{f}(\bar{x}_j\mathbb{1}^\mathsf{T})\mathbb{1}_u\mathbb{1}^\mathsf{T} - z_j z_j^\mathsf{T}\nabla\boldsymbol{f}(\boldsymbol{x}_j)\mathbb{1}_u\mathbb{1}^\mathsf{T}\|^2_F \\
& \leq 8\|z_j z_j^\mathsf{T}\nabla\boldsymbol{f}(\boldsymbol{x}_j) - z_j z_j^\mathsf{T}\nabla\boldsymbol{f}(\bar{x}_j\mathbb{1}^\mathsf{T})\|^2_F + 2\|z_j z_j^\mathsf{T}\nabla\boldsymbol{f}(\bar{x}_j\mathbb{1}^\mathsf{T}) - z_j z_j^\mathsf{T}\nabla F(\bar{x}_j\mathbb{1}^\mathsf{T})\mathbb{1}^\mathsf{T}\|^2_F
\end{aligned}
\tag{71}
$$

where we utilize the identity that $\nabla\boldsymbol{F}(\bar{x}_j\mathbb{1}^\mathsf{T}) = \nabla\boldsymbol{f}(\bar{x}_j\mathbb{1}^\mathsf{T})\mathbb{1}_u$. Recall that

$$
\mathbb{E}\, z_j z_j^\mathsf{T} z_j z_j^\mathsf{T} = \mathrm{Tr}(H_r^{-1})H_r^{-1} + 2H_r^{-2} := \Phi_r H_r^{-1}
\tag{72}
$$

where $r$ is the corresponding round for the iteration $j$. Notice $\|\Phi_r\| \leq (d+2)/\beta_\ell$, which is not a tight bound though. Hence, taking the expectation with respect to $z_j$, we establish

$$
\mathbb{E}\,\|\boldsymbol{x}_{k+1} - \bar{x}_{k+1}\mathbb{1}^\mathsf{T}\|^2_F
$$

$$\leq 8\eta^2(\tau-1)\sum_{j=r\tau+1}^{k}\|\nabla \boldsymbol{f}(\boldsymbol{x}_j)-\nabla \boldsymbol{f}(\bar{x}_j\mathbb{1}^\mathsf{T})\|^2_{\Phi_r H_r^{-1}}$$

$$+2\eta^2(\tau-1)\sum_{j=r\tau+1}^{k}\|\nabla \boldsymbol{f}(\bar{x}_j\mathbb{1}^\mathsf{T})-\nabla F(\bar{x}_j\mathbb{1}^\mathsf{T})\mathbb{1}^\mathsf{T}\|^2_{\Phi_r H_r^{-1}}+O(\eta^2\mu^2)$$

$$\leq 8\eta^2(\tau-1)L\beta_\ell^{-1}\|\Phi_r\|\sum_{j=r\tau+1}^{k}\|\boldsymbol{x}_j-\bar{x}_j\mathbb{1}^\mathsf{T}\|^2_F+2\eta^2(\tau-1)^2 M\beta_\ell^{-1}\|\Phi_r\|(\sigma_G^2+\sigma_s^2)+O(\eta^2\mu^2)$$

$$(73)$$

Lastly, we just need to take another summation over $k$ from $r\tau$ to $(r+1)\tau-2$. Recall that $\|\boldsymbol{x}_{r\tau+1}-\bar{x}_{r\tau+1}\mathbb{1}^\mathsf{T}\|^2_F=0$. After rearranging and utilizing the fact that $\sum_{k=r\tau}^{(r+1)\tau-2}\sum_{j=r\tau+1}^{k}a_j\leq(\tau-1)\sum_{k=r\tau+1}^{(r+1)\tau}a_k$ for any nonnegative value $a_k$, we have

$$\left(1-8\eta^2(\tau-1)^2 L\beta_\ell^{-1}\|\Phi_r\|\right)\frac{1}{\tau}\sum_{k=r\tau+1}^{(r+1)\tau}\|\mathbb{E}\|\boldsymbol{x}_k-\bar{x}_k\mathbb{1}^\mathsf{T}\|^2_F$$

$$\leq 2\eta^2(\tau-1)^2 M\beta_\ell^{-1}\|\Phi_r\|(\sigma_G^2+\sigma_s^2)+O(\eta^2\mu^2) \qquad (74)$$

After restricting $\eta$ to force $1-8\eta^2(\tau-1)^2 L\beta_\ell^{-1}\|\Phi_r\|<1/2$, we establish this lemma. $\qquad\square$

A special case is local update step $\tau=1$. In this case, we do not need any consensus error since the models are all synchronized. We can simply discard the term $\mathbb{E}\|\boldsymbol{x}_k-\bar{x}_k\mathbb{1}^\mathsf{T}\|^2_F$ in the descent lemma.

### F.6 CONVERGENCE ANALYSIS OF THEOREM 1

#### F.6.1 CONVERGENCE PROOF OF THEOREM 1

**Proof:** We are now ready to present the convergence theorem, which simply combines the consensus lemma and the descent lemma above then taking the double exeception.

$$\mathbb{E}\left[F(\bar{x}_{(r+1)\tau+1})\right]\leq\mathbb{E}\left[F(\bar{x}_{r\tau+1})\right]-\frac{\eta}{4}\sum_{j=r\tau}^{(r+1)\tau-1}\mathbb{E}\|\nabla F(\bar{x}_j)\|^2_{H_r^{-1}}+O(\eta^2\mu)$$

$$+\frac{4\tau\eta^2\rho_k}{\beta_\ell m}(\sigma_G^2+\sigma_s^2)+\frac{8\eta^2(\tau-1)^2 L}{\tau m}\sum_{j=r\tau}^{(r+1)\tau-1}\|\Phi_r\|(\sigma_G^2+\sigma_s^2) \qquad (75)$$

Expanding the summations and re-arranging terms, we obtain

$$\frac{1}{\tau R}\sum_{j=1}^{\tau R}\mathbb{E}\|\nabla F(\bar{x}_j)\|^2_{H_r^{-1}}\leq\frac{4(F(\bar{x}_1)-F^\star)}{\eta\tau R}+\frac{16\eta\bar{\rho}}{\beta_\ell m}(\sigma_G^2+\sigma_s^2)+\frac{32\eta(\tau-1)^2 L\bar{\phi}}{\beta_\ell\tau m}(\sigma_G^2+\sigma_s^2)$$

$$+\mathcal{O}(\eta\mu), \qquad (76)$$

where

$$\bar{\rho}=\frac{1}{K}\sum_{k=0}^{K}\rho_k=\frac{1}{K}\sum_{k=0}^{K}(\text{Tr}(\Xi_k)+2\|\Xi_k\|) \qquad (77)$$

$$=\frac{1}{K}\sum_{k=0}^{K}(\text{Tr}(H_k^{-1/2}\Sigma_k H_k^{-1/2})+2\|H_k^{-1/2}\Sigma_k H_k^{-1/2}\|) \qquad (78)$$

$$\bar{\phi}=\frac{1}{R}\sum_{r}\|\Phi_r\|=\frac{1}{R}\sum_{r}(\text{Tr}(H_r^{-1})+2\|H_r^{-1}\|) \qquad (79)$$

Combining all learning rate requirements, we have

$$\eta\leq\min\left(\frac{\beta_\ell}{mL},\frac{1}{8\rho_k},\frac{\beta_\ell}{4(\tau-1)}\sqrt{\frac{1}{L(d+2)}}\right) \qquad (80)$$

Lastly, translating the above result back to the two-level $k$ and $r$ indexing, we establish Theorem 1.

### F.6.2 CONVERGENCE RATE

To establish the convergence rate, we distinguish two scenarios – the local update $\tau = 1$ and the local update $\tau > 1$. When $\tau = 1$, the rate becomes much simpler

$$\frac{1}{R}\sum_{r=0}^{R-1}\mathbb{E}\left\|\nabla F(\bar{x}_{r,0})\right\|_{H_r^{-1}}^2 \leq \frac{4(F(\bar{x}_1) - F^\star)}{\eta R} + \frac{16\eta\bar{\rho}}{\beta_\ell m}(\sigma_G^2 + \sigma_s^2) + \mathcal{O}(\eta\mu), \tag{81}$$

When the communication round $R$ is sufficiently large and the ZO smoothing parameter $\mu$ is sufficiently small, we choose the learning rate $\eta = \sqrt{\frac{m\beta_\ell}{\bar{\rho}R}}$, which leads to the following rate:

$$\frac{1}{R}\sum_{r=0}^{R-1}\mathbb{E}\left\|\nabla F(\bar{x}_{r,0})\right\|_{H_r^{-1}}^2 = \mathcal{O}\left(\sqrt{\frac{\bar{\rho}}{mR}}\right) \tag{82}$$

Based on the Table 1, we can establish the following four rates based on the conditions:

1. $H_r$ is a well-approximated one with $L$-smoothness assumption, then the rate is $\mathcal{O}\left(\sqrt{\frac{d}{mR}}\right)$.

2. $H_r$ is a well-approximated one with low effective rank, then the rate is $\mathcal{O}\left(\sqrt{\frac{\zeta}{mR}}\right)$.

3. DeComFL Case: No Hessian information is learned, i.e., $H_k \equiv I$, with $L$-smoothness assumption, then the rate is $\mathcal{O}\left(\sqrt{\frac{Ld}{mR}}\right)$.

4. DeComFL Case: No Hessian information is learned, i.e., $H_k \equiv I$, with low effective rank, then the rate is $\mathcal{O}\left(\sqrt{\frac{L\kappa}{mR}}\right)$.

For the local update $\tau > 1$ case, we choose the learning rate $\eta = \min\left(\sqrt{\frac{m\beta_\ell}{\tau\bar{\rho}R}}, \sqrt{\frac{m\beta_\ell}{\tau\bar{\phi}R}}\right)$. Then we obtain the following rate

$$\frac{1}{\tau R}\sum_{r=0}^{R-1}\sum_{k=0}^{\tau-1}\mathbb{E}\left\|\nabla F(\bar{x}_{r,k})\right\|_{H_r^{-1}}^2 = \underbrace{\mathcal{O}\left(\sqrt{\frac{\bar{\rho}}{\tau mR}}\right)}_{\text{descent residue}} + \underbrace{\mathcal{O}\left(\sqrt{\frac{\tau\bar{\phi}}{mR}}\right)}_{\text{consensus residue}} \tag{83}$$

where the second extra term comes from the client model diverging in the local update steps.

Similarly, we can establish the four rates based on the assumption. Here we focus on the low effective rank case since it reveals the difference between DeComFL and `HiSo`.

When $H_r \equiv I$, we have $\bar{\phi} = d + 2$ and $\bar{\rho} \leq L\kappa$. Therefore, we establish the following rate for DeComFL rate:

$$\mathcal{O}\left(\sqrt{\frac{L\kappa}{\tau mR}}\right) + \mathcal{O}\left(\sqrt{\frac{\tau d}{mR}}\right) \tag{84}$$

Here we can see that even if $\bar{\rho}$ can be tighter bounded by low-effective rank, the convergence rate still depends on $d$.

In contrast, if $H_r$ well-approximates the Hessian $\Sigma$ with the low effective rank, we establish the convergence rate for `HiSo` is

$$\mathcal{O}\left(\sqrt{\frac{\zeta}{\tau mR}}\right) + \mathcal{O}\left(\sqrt{\frac{\tau\kappa}{mR}}\right) \tag{85}$$

Now, if we compare Eq. (84) with Eq. (85), we can tell that `HiSo` is still capable of being independent of Lipschitz $L$ and model dimension $d$; meanwhile, DeComFL cannot. This probably explains why the original paper (Li et al., 2025b) cannot provide the proof for the dimension-free rate with $\tau > 1$. Of course, Eq. (84) is just an upper bound for the worst-case scenario. The practical performance may not be pessimistic as the bound indicates.

### F.7 More Well-approximation of Hessian Matrix Analysis and Experiments

### F.7.1 Theoretical Analysis When Well-approximation Does not Hold

As mentioned in the main context, whether this approximation holds in the context of LLMs is difficult to assess. To understand the impact of the approximation of the Hessian, we refer back to the key quantity in Theorem 1.

$$\bar{\rho} = \frac{1}{\tau R} \sum_r \sum_k (\text{Tr}(H_r^{-1/2} \Sigma_{r,k} H_r^{-1/2}) + 2\|H_r^{-1/2} \Sigma_{r,k} H_r^{-1/2}\|). \tag{86}$$

This is the dominant and distinguished term compared with other zeroth-order methods. Among these two terms in the double summation, $\text{Tr}(H_r^{-1/2} \Sigma_{r,k} H_r^{-1/2})$ is usually much larger than $\|H_r^{-1/2} \Sigma_{r,k} H_r^{-1/2}\|$. Hence, we can just focus on the former term only. Because both $\Sigma_{r,k}$ and $H_r$ are positive semi-definite matrices, we know

$$\sum_{j=1}^d \lambda_j(\Sigma_{r,k})/\lambda_k(H_r) \leq \text{Tr}(H_r^{-1/2} \Sigma_{r,k} H_r^{-1/2}) \leq \sum_{j=1}^d \lambda_j(\Sigma_{r,k})/\lambda_{-k}(H_r), \tag{87}$$

where the notation $\lambda_j(\cdot)$ denotes the $j$-th eigenvalue of a matrix, ranked in descending order (from largest to smallest), and $\lambda_{-j}(\cdot)$ represents the eigenvalues in ascending (reverse) order.

At the early stage $H_r \approx I$, then $\text{Tr}(H_r^{-1/2} \Sigma_{r,k} H_r^{-1/2}) \approx \text{Tr}(\Sigma_{r,k})$, equivalent to the no learning of Hessian. This can be thought as the baseline. Now, suppose we unfortunately learn a poor Hessian approximation that amplifies the largest eigenvalues of the Hessian while shrinking the smallest ones. In this case, we will obtain a larger $\rho_{r,k}$ than $\text{Tr}(\Sigma_{r,k})$, resulting in a worse outcome than the baseline that does not apply Hessian information. However, this case is unlikely to be stable, since $H_r$ will gradually improve its approximation of $\Sigma_{r,k}$ to some degree. Most commonly, the value of $\rho_{r,k}$ should lie between the bounds of equation 87. $H_r$ will not have the perfect same eigenspace as $\Sigma_{r,k}$. But as long as the largest few eigenvalues of $\Sigma_{r,k}$ are divided by correspondingly large values from $H_r$, the total sum should be smaller than $\text{Tr}(\Sigma_{r,k})$. This is highly probable due to the long-tail distribution of the Hessian's eigenvalues.

Lastly, notice $\bar{\rho}$ has an $O(1/R)$ decay rate, governing how quickly old values are forgotten. It is common for the estimation to be inaccurate at the beginning, but it eventually converges to a stable value. This $O(1/R)$ rate matches the algorithm's convergence rate. Hence, as long as the Hessian approximation converges no slower than $O(1/R)$, any early, inaccurate estimates will not negatively impact the final rate.

### F.7.2 Estimate the impact of meaningful Hessian Approximation

In Figure 5, we demonstrated that the final performance is relatively insensitive to the choice of $\nu$, provided that $\nu$ remains close to 1. This is because high values of $\nu$ ensure sufficient smoothing of the Hessian estimate. To further investigate the sensitivity, we conducted additional experiments using the same setup but with significantly smaller values of $\nu$ (e.g., $\nu < 0.5$). In this regime, the estimator effectively utilizes only the most recent 1-2 rounds of zeroth-order information, resulting in a high-variance and uninformative Hessian approximation. The results, shown in Figure 9, reveal that performance deteriorates as $\nu$ decreases, confirming that a well-accumulated Hessian estimate is crucial for acceleration. Importantly, while a poor Hessian approximation degrades performance, the algorithm still converges to a reasonable solution. Since the preconditioner $H$ remains a positive definite diagonal matrix, the "worst-case" scenario resembles a poorly scaled first-order method rather than a catastrophic failure. Conversely, these results validate that when the Hessian is well-learned (high $\nu$), it provides meaningful acceleration.

### F.7.3 Hessian Approximation Evaluation in Random Subspaces

While exact computation for the full model is prohibitive, we can rigorously analyze the exact Hessian within restricted subspaces. To do so, we randomly sampled 1000 parameters to compute the exact $1000 \times 1000$ Hessian matrix $\hat{\Sigma}$. We then projected the learned Hessian $H$ from HiSo into this same subspace to obtain $\hat{H}$.

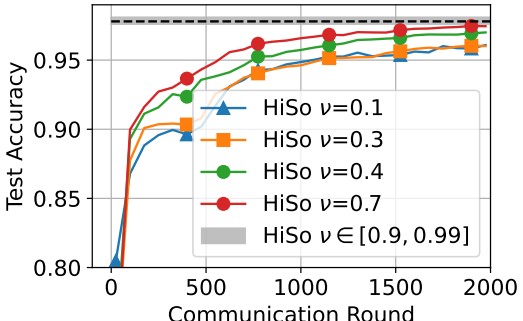

Figure 9: Impact of smoothing parameter $\nu$ when $\nu$ is way smaller than 0.9. The gray area indicates the final performance of HiSo when $\nu$ is chosen in $[0.9, 0.99]$ range. We can view the $\nu = 0.1$ case does not learn a good Hessian approximate, hence the gap between the curves indicates improvement of good Hessian approximation.

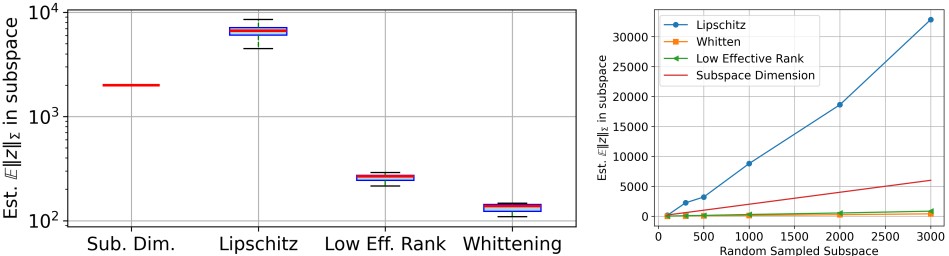

Figure 10: Estimation of $\mathbb{E}\|z\|_\Sigma^2$ upper bounds in a restricted subspace. We randomly sampled 1,000 parameter indices from the final trained model in Figure 5 to extract the exact subspace Hessian and the corresponding learned diagonal Hessian $H$. The plotted quantities correspond to the four upper bounds listed in Table 1: "Sub. Dim." ($2d$), "Lipschitz" ($Ld$, estimated via the largest eigenvalue of the subspace Hessian), "Low Eff. Rank" ($L\kappa$), and "Whitening" ($\zeta$). The left boxplot figure reports the values estimated across 10 independently sampled subspaces. The right figure illustrates how these quantities vary as the dimension of the sampled subspace increases.

The eigenvalue correlation between $\hat{\Sigma}$ and $\hat{H}$ is modest (7-20%, which we consider a reasonable value for diagonal approximations). But when we evaluate the quantity $\mathbb{E}\|\hat{u}\|_{\hat{\Sigma}}$ discussed in Table 1, we observe much better improvements. The whitened Hessian eigenvalues are significantly smaller than the dimension $d$ and lower than the low effective rank estimation in that subspace. Specifically, we observe $\hat{\zeta} \sim 150$, the estimated $\hat{L}\kappa \sim 250$, and $\hat{L}d \sim 5000$. Compared to the ideal synthetic result in figure 4, it is reasonable to expect that the improvement is not significant. But this improvement is more or less consistent with our LLM experiment in Figure 7. We repeated this process across multiple sampled subspaces; the boxplot in the left one Figure 10 confirms the consistency of this observation.

Finally, we acknowledge that subspace Hessian eigenvalues are not unbiased estimators of the full Hessian. To mitigate this gap, we plotted the estimated quantities across varying subspace dimensions (the right one in Figure 10). As expected, the worst-case Lipschitz estimation $Ld$ increases rapidly with dimension, whereas the whitening metric remains robust and insensitive to changes in dimension.

## G   MULTI-PERTURBATION VERSION

Following our detailed examination of ZO-gradient variance, it is evident that reducing this variance is crucial for enhancing the performance of ZO-based methods. In this context, **multi-perturbation sampling in ZO-SGD can be viewed as analogous to mini-batching in standard SGD**, where multiple samples are used to improve the quality of the gradient estimate.

In terms of `HiSo`, the multi-perturbation version is simply replacing the finding $\Delta x_{r,k}^{(i)}$ step by the following:

$$
\begin{aligned}
&\text{for } p = 0, 1 \cdots, P - 1 : \\
&\quad u_{r,k,p} \sim \mathcal{N}(0, I) \\
&\quad g_{r,k,p}^{(i)} = \frac{1}{\mu}[f_i(x_{r,k}^{(i)} + \mu H_r^{-1/2} u_{r,k,p}) - f_i(x_{r,k}^{(i)})] \\
&\Delta x_{r,k}^{(i)} = H_r^{-1/2} \frac{1}{P} \sum_{p=0}^{P-1} g_{r,k,p}^{(i)} u_{r,k,p}
\end{aligned}
\tag{88}
$$

Notice for the multi-perturbation version, we need to transmit $P$ random seeds to generate $p$ random vector $u_{r,k,p}$. Moreover, $P$ local gradient scalars $g_{r,k,p}^{(i)}$ are required to be communicated as well.

At the server side, the aggregation step now is required to average $P$ values separately:

$$
\Delta x_{r,k} = \frac{1}{\tau|C_r|} \sum_{i \in C_r} \sum_{k=0}^{\tau-1} \Delta x_{r,k}^{(i)} = \frac{1}{\tau} \sum_{k=0}^{\tau-1} \left[ \frac{1}{P} \sum_{p=0}^{P-1} \underbrace{\left( \frac{1}{|C_r|} \sum_{i \in C_r} g_{r,k,p}^{(i)} \right)}_{:=g_{r,k,p}} H_r^{-1/2} u_{r,k,p} \right]
\tag{89}
$$

Notice we can switch the order of summation in above equations because $u_{r,k,p}$ is common among all clients. This aggregated gradient scalar $g_{r,k,p}$ stands for the $r$-th round, $k$-th local update, and $p$-th perturbation. $P$ gradient scalars together with $P$ random seeds are sufficient to reconstruct the global $\Delta x_{r,k}$. For the reconstruction step, everything is the same.

## G.1 PERFORMANCE ANALYSIS

**Theorem 2** (Multi-Perturbation Version). *Under Assumptions 1, 2, 3 and 4, if $\eta \leq \min\left( \frac{\beta_\ell}{mL}, \frac{1}{8\rho_{k,P}}, \frac{\beta_\ell}{4(\tau-1)} \sqrt{\frac{1}{L(d+2)}} \right)$, the sequence of iterates generated by `HiSo` with $P$ perturbations satisfies:*

$$
\frac{1}{\tau R} \sum_{r=0}^{R-1} \sum_{k=0}^{\tau-1} \mathbb{E} \|\nabla F(\bar{x}_{r,k})\|_{H_r^{-1}}^2 \leq \frac{4(F(\bar{x}_1) - F^\star)}{\eta \tau R} + \underbrace{\frac{32\eta(\tau-1)^2 L\bar{\phi}_P}{\beta_\ell \tau m}(\sigma_G^2 + \sigma_s^2)}_{\text{extra client drift term}} + \frac{16\eta\bar{\rho}_P}{\beta_\ell m}(\sigma_G^2 + \sigma_s^2)
$$
$$
+ \mathcal{O}(\eta\mu),
\tag{90}
$$

*where*

$$
\bar{\rho}_P = \frac{1}{\tau R} \sum_r \sum_k \left( \frac{1}{P} \operatorname{Tr}(H_r^{-1/2}\Sigma_{r,k}H_r^{-1/2}) + (\frac{1}{P} + 1)\|H_r^{-1/2}\Sigma_{r,k}H_r^{-1/2}\| \right)
\tag{91}
$$

$$
\bar{\phi}_P = \frac{1}{R} \sum_r \left( \frac{1}{P} \operatorname{Tr}(H_r^{-1}) + (\frac{1}{P} + 1)\|H_r^{-1}\| \right)
\tag{92}
$$

*and the rest of the quantities are the same as Theorem 1.* □

**Proof**: In this case, the algorithm formulation can be written as

$$
y_{k+1} = x_k - \eta\frac{1}{P} \sum_{p=1}^{P} z_{k,p}z_{k,p}^\mathsf{T}\nabla f(x_k; \xi_k) + O(\mu\eta),
\tag{93}
$$

$$
x_{k+1} = y_{k+1}W_k,
\tag{94}
$$

Notice there are three sources of the randomness – random direction $z$, gradient noise coming from $\xi_k$m and the sampling randomness $W_k$. They are independent of each other, so we can treat them one by one separately. It is straightforward to verify that the mean is unchanged

$$
\mathbb{E}\frac{1}{P} \sum_{p=1}^{P} z_{k,p}z_{k,p}^\mathsf{T}\nabla f(x_k; \xi_k) = H_k^{-1}\nabla f(x_k)
\tag{95}
$$

Next, noting $\{z_{k,p}\}_p$ is independent and identically distributed, utilizing lemma 1 we establish

$$\frac{1}{P^2} \sum_{p'=1}^{P} \sum_{p=1}^{P} \mathbb{E}\, z_{k,p} z_{k,p}^{\mathsf{T}} \Sigma_k z_{k,p'} z_{k,p'}^{\mathsf{T}}$$

$$= \frac{P^2 - P}{P^2} H_k^{-1} \Sigma_k H_k^{-1} + \frac{1}{P^2} \sum_{p=1}^{P} \mathbb{E}\, z_{k,p} z_{k,p}^{\mathsf{T}} \Sigma_k z_{k,p} z_{k,p}^{\mathsf{T}}$$

$$= \frac{P-1}{P} H_k^{-1} \Sigma_k H_k^{-1} + \frac{1}{P} (\mathrm{Tr}(\Sigma_k H_k^{-1}) H_k^{-1} + 2 H_k^{-1} \Sigma_k H_k^{-1})$$

$$= \frac{1}{P} \mathrm{Tr}(\Sigma_k H_k^{-1}) H_k^{-1} + \left( \frac{1}{P} + 1 \right) H_k^{-1} \Sigma_k H_k^{-1} \tag{96}$$

Recall that this quantity $\rho_k$ of the single perturbation case is

$$\rho_k = \mathrm{Tr}(H_k^{-1/2} \Sigma_k H_k^{-1/2}) + 2\|H_k^{-1/2} \Sigma_k H_k^{-1/2}\|^2$$

The multi-perturbation version one will become

$$\rho_{k,P} = \frac{1}{P} \mathrm{Tr}(H_k^{-1/2} \Sigma_k H_k^{-1/2}) + \left( \frac{1}{P} + 1 \right) \|H_k^{-1/2} \Sigma_k H_k^{-1/2}\|^2 \approx \frac{1}{P} \rho_k$$

Recall that the first term in $\rho_k$ is typically much bigger than the second one. Hence, $\rho_{k,P} \approx \rho_k/P$ as we expect that multi-perturbation will decrease the variance of the random search direction.

Besides, it is a similar case applied to quantity:

$$\frac{1}{P^2} \sum_{p'=1}^{P} \sum_{p=1}^{P} \mathbb{E}\, z_{k,p} z_{k,p}^{\mathsf{T}} z_{k,p'} z_{k,p'}^{\mathsf{T}} = \frac{1}{P} \mathrm{Tr}(H_k^{-1}) H_k^{-1} + \left( \frac{1}{P} + 1 \right) H_k^{-1} H_k^{-1} \tag{97}$$

So that the multi-perturbation version of $\phi_{r,P}$ will become

$$\phi_{r,P} = \frac{1}{P} \mathrm{Tr}(H_r^{-1}) + \left( \frac{1}{P} + 1 \right) \|H_r^{-1}\|^2 \approx \frac{1}{P} \phi_r$$

Notice we just need to update the Eq. (58) with the result of Eq. (96). After some calculations and simplification, we establish the result of Theorem 2.

## G.2    CONVERGENCE RATE

Notice the relationship $\rho_{k,P} \approx \rho_k/P$, we can immediately establish that for $\tau = 1$ the convergence rate of `HiSo` is $\mathcal{O}\left(\sqrt{\frac{\bar{\rho}_P}{mR}}\right)$. Further, under the well-approximated Hessian assumption, we can establish the dimension-free rate

$$\frac{1}{R} \sum_{r=0}^{R-1} \|\nabla F(\bar{x}_{r,0})\|_{H_r^{-1}}^2 = \mathcal{O}\left(\sqrt{\frac{\zeta}{mPR}}\right) \tag{98}$$

When $\tau > 1$, we have $\mathcal{O}\left(\sqrt{\frac{\bar{\rho}}{\tau mR}}\right) + \mathcal{O}\left(\sqrt{\frac{\tau \bar{\phi}}{mR}}\right)$. Further, under the well-approximated Hessian assumption, we can establish the dimension-free rate

$$\frac{1}{\tau R} \sum_{r=0}^{R-1} \sum_{k=0}^{\tau-1} \|\nabla F(\bar{x}_{r,k})\|_{H_r^{-1}}^2 = \mathcal{O}\left(\sqrt{\frac{\zeta}{\tau mPR}}\right) + \mathcal{O}\left(\sqrt{\frac{\tau \kappa}{mPR}}\right) \tag{99}$$

