# OpenReview forum: "Converge Faster, Talk Less: Hessian-Informed Federated Zeroth-Order Optimization"
_ICLR.cc/2026/Conference — ICLR 2026 Poster_

### Official Review · Reviewer_7YFj · 2025-10-30

**Soundness:** 3
**Presentation:** 3
**Contribution:** 4
**Rating:** 8
**Confidence:** 4

**Summary:**

This paper proposes HiSo, a novel Hessian-informed zeroth-order optimization method for federated learning that achieves dimension-free communication while accelerating convergence. The key innovation lies in leveraging global diagonal Hessian approximations as preconditioners without transmitting second-order information, thus preserving the scalar-only communication paradigm. Theoretically, the authors establish a convergence rate independent of model dimension and Lipschitz constant under certain Hessian approximation assumptions, providing the first such result for ZO methods in FL with multiple local updates. Empirically, HiSo demonstrates 1-5× speedup in communication rounds and up to 90 million times communication savings compared to first-order baselines across diverse LLM fine-tuning tasks.

**Strengths:**

The primary strength is the elegant resolution of a fundamental tension between leveraging second-order information for acceleration and maintaining extreme communication efficiency. HiSo represents a conceptually novel approach with non-trivial theoretical analysis and compelling empirical results. The scalar-only communication property holds significant practical value for bandwidth-constrained federated LLM fine-tuning.

**Weaknesses:**

The main weakness concerns the foundational assumptions underlying its theoretical advantages. The critical "well-approximated Hessian" condition is challenging to verify empirically, particularly during early training stages. While outperforming ZO baselines, the accuracy gap with first-order methods remains, highlighting inherent limitations of the ZO approach.

**Questions:**

The theoretical advantages rely on the "well-approximated Hessian" condition. Do you have any empirical evidence (e.g., comparing with true Hessian on small models) suggesting your learning method effectively captures principal curvature directions?

In extreme FL scenarios with massive client populations (e.g., millions), could maintaining client participation history become a server-side memory bottleneck? How do you assess this scalability aspect?

The Hessian smoothing parameter ν appears to have minimal impact. Does this indicate HiSo is genuinely insensitive to this hyperparameter, or are there implicit guidelines for its selection?

---

> ### Author Response · Authors · 2025-11-19
> **Response for W1&Q1, Q2, Q3**
>
> ### **W1&Q1: About "well-approximated Hessian" and the accuracy gap between first-order (FO) and ZO methods.**
>
> **Our response:** Thank you so much for your comment, and we would like to address the two parts below.
> (1) Since several reviewers raised questions regarding the "well-approximated Hessian," we would like to provide a unified clarification about this concern. Please refer to our global response for details.
>
> (2) Regarding the accuracy gap between ZO and first-order methods, we acknowledge that this is an inherent property of ZO optimization. ZO methods rely on stochastic directional estimates and therefore naturally exhibit higher variance than first-order gradients. The goal of this paper is to design a ZO method with faster convergence under scalar-only communication constraints, and in our experiment evaluation, HiSo consistently outperforms existing ZO baselines on final test accuracy, communication cost, required rounds.
>
> ---
>
> ### **Q2: About memory cost for maintaining client participation history.**
>
> **Our response:** Thank you for this important question. As shown in the memory-usage analysis in Appendix E.2 (b) of the original paper, the server-side cost of maintaining client participation history is extremely small. Even in a FL system with one million clients, storing one million 32-bit integers requires only 3.8 MB, which is negligible for modern server infrastructures. Therefore, the memory footprint remains minimal and highly scalable, even in very large-scale FL deployments.
>
> ---
>
> ### **Q3: About parameter $\nu$ and its selection.**
>
> **Our response:** Thank you for the question. To be precise, HiSo is not strictly insensitive to the Hessian smoothing parameter $\nu$; rather, it is robust within the recommended range of $\nu \in [0.9, 1)$. In this interval, the smoothing factor allows for sufficient accumulation of curvature information, yielding a stable diagonal Hessian approximation. Consequently, performance is consistently strong, and we suggest selecting $\nu$ from this range.
>
> To illustrate the sensitivity outside this range, we added a new experiment (Appendix F.7.2 and Figure 9) using significantly smaller values (e.g., $\nu < 0.5$). In this regime, the estimator’s effective memory is reduced to only the most recent 1–2 rounds, resulting in a high-variance and uninformative Hessian approximation. As expected, our results show that performance deteriorates as $\nu$ decreases.
>
> In summary, the performance appears insensitive when $\nu$ is close to 1 because the approximation quality saturates at a "good enough" level. However, the degradation observed at small $\nu$ confirms that the acceleration is indeed driven by the quality of the accumulated Hessian estimate.

---

### Official Review · Reviewer_3tFi · 2025-11-01

**Soundness:** 2
**Presentation:** 3
**Contribution:** 2
**Rating:** 4
**Confidence:** 3

**Summary:**

This paper proposes HiSo, a novel federated learning (FL) algorithm designed for communication-efficient fine-tuning of Large Language Models (LLMs). HiSo innovatively combines zeroth-order (ZO) optimization for dimension-free communication with Hessian-informed preconditioning to accelerate convergence. The core idea is to use a global, diagonal Hessian approximation to guide the random search directions in ZO optimization, effectively creating a "natural gradient" style update. Crucially, this Hessian information is learned and applied without transmitting any second-order matrices, preserving the scalar-only communication property that makes ZO methods attractive for high-dimensional problems. Theoretically, the authors provide a convergence analysis suggesting that under a "low-effective-rank" assumption for the Hessian, HiSo can achieve a rate independent of the model dimension d and Lipschitz constant L. Empirically, HiSo demonstrates significant speedups (1-5x in communication rounds) and up to 90 million times lower communication cost compared to first-order baselines like FedAvg on LLM fine-tuning tasks.

**Strengths:**

1.  Novel and Well-Motivated Idea: The combination of Hessian-informed updates with scalar-only communication is highly innovative. It directly addresses the primary weakness of ZO methods (slow convergence) while preserving their key advantage (low communication cost). The motivation is clear and grounded in the practical challenge of federated LLM fine-tuning.
2.  Strong Empirical Evaluation: The experiments are comprehensive, spanning from simple CNN/MNIST setups to LLM fine-tuning on standard NLP benchmarks (SST-2, QQP, SQuAD). The comparisons against a wide range of baselines (FedAvg, FedAdam, FedZO, DeComFL) are convincing and demonstrate clear improvements in convergence speed and final accuracy for ZO methods.
3.  Significant Practical Impact: The reported communication savings (MBs vs TBs) are monumental. If applicable to even larger models, HiSo could have a substantial impact on the feasibility of federated fine-tuning in bandwidth-constrained environments.
4.  Theoretical Depth: The paper goes beyond mere algorithm design by providing a non-trivial theoretical analysis. The introduction of the "low whitening rank"  to explain the accelerated convergence is a valuable conceptual contribution that helps reconcile the practical efficiency of ZO methods with their pessimistic worst-case theoretical bounds.

**Weaknesses:**

1.  Limited Discussion on Computation Overhead: While communication is the primary bottleneck, the computational cost of ZO methods is inherently higher than first-order methods due to the need for multiple forward passes. The paper briefly mentions the preconditioning time is negligible but does not provide a full comparison of the total wall-clock time (computation + communication) against first-order methods. This is crucial for assessing real-world utility, as increased computation time might offset communication savings.
2.  Validation of Theoretical Assumptions: The core theoretical improvement hinges on the "well-approximated Hessian" condition (Eq. 17) and the low-effective-rank property. While the empirical results are strong evidence, the paper does not directly validate that the learned diagonal matrix H actually satisfies this condition for the LLMs used in the experiments. A more rigorous analysis or measurement of the whitening rank  during training would strengthen the theoretical claims.
3.  Comparison with Parameter-Efficient Fine-Tuning (PEFT): The discussion of FL+PEFT baselines (like FedLoRA) is brief and relegated to the appendix. Given that PEFT is a dominant approach for efficient LLM fine-tuning, a more thorough comparison in the main text is warranted. HiSo's claim of "full-parameter" tuning is a different paradigm, but a direct comparison on metrics like final accuracy, communication cost, and memory usage would better situate HiSo within the existing landscape.
4.  Hyperparameter Sensitivity: The ablation study on the Hessian smoothing parameter ν shows robustness, but the performance of adaptive methods like HiSo can be sensitive to other hyperparameters like the learning rate and the exponential moving average decay factor. A more detailed sensitivity analysis would be helpful for practitioners.

**Questions:**

1.  How does the total energy consumption (a function of both communication and computation) of HiSo compare to first-order methods and DeComFL, especially when considering the additional forward passes required for ZO estimation?
2.  Could the HiSo framework be naturally extended to incorporate PEFT techniques (e.g., applying Hessian-informed updates only to low-rank LoRA parameters)? Do you see this as a promising future direction?
3.  The theory suggests performance degrades to match DeComFL if the Hessian approximation is poor. Did you observe this in practice during the initial stages of training or on any specific tasks?

---

> ### Author Response · Authors · 2025-11-19
> **Response for W1&Q1, W2**
>
> ### **W1&Q1: About computation cost.**
>
> **Our response:** Thank you for your comments and question. In the Table 1 below, we report the one-round computation time, one-round communication time and total wall-clock time until convergence of first-order baselines, zeroth-order baseline and our HiSo. We observe that first-order FedAvg and FedAdam suffer from extremely high wall-clock time due to their high-dimensional transmission overhead that each round requires $228.8$ seconds of communication, dominating the overall runtime. Thus, despite their low per-round computation time, they require $68,788$ s and $45,858$ s total wall-clock time respectively.
>
> In contrast, zeroth-order DeComFL significantly reduce communication to only $2.24\times 10^{-6}$ seconds per round, achieving a $29\times$ faster overall runtime than FedAdam. However, DeComFL still requires many more communication rounds to reach the target accuracy, compared to other three methods.
>
> Further, our HiSo achieves the best overall efficiency. Although the per-round computation time is slightly higher due to Hessian-informed updates, HiSo converges in far fewer rounds compared to DeComFL. Hence, HiSo achieves a total wall-clock time of only $1064$ seconds, representing a $43\times$ speedup over FedAdam, $64\times$ speedup over FedAvg, and a $2.2\times$ improvement over DeComFL. This shows that HiSo not only preserves the ultra-low communication cost of scalar-only communication but also substantially accelerates convergence (i.e., less training rounds) through its Hessian-informed optimization design.
>
> Table 1: Time Cost Comparison (Qwen3-0.6B Model+SST-2 Dataset). We assume a typical 5G network setting with 100 Mbps uplink and 500 Mbps downlink bandwidth without considering latency, protocol and so on. We tested this on H100 GPUs.
> |Method|One-round computation time|One-round Communication time|Total wall-clock time until convergence|
> |-|-|-|-|
> |FedAvg|$0.43$ s|$228.8$ s|$68788$ s|
> |FedAdam|$0.49$ s|$228.8$ s|$45858$ s
> |DeComFL (P=5)|$0.94$ s|$2.24\times 10^{-6}$ s|$2350$ s|
> |HiSo (P=5)|$1.12$ s|$2.24\times 10^{-6}$ s|$1064$ s|
>
> If we consider using PCIe 4.0 to transfer models between clients, the typical speed of PCIe 4.0 is about 2.5 GB/s. Then, we can estimate the wall-clock time under this scenario by proportionally scaling the communication time.
>
> ---
>
> ### **W2: Validation of Theoretical Assumptions.**
>
> **Our response:** Thank you so much for your comment. We acknowledge the importance of rigorously analyzing the proposed condition. However, computing the full Hessian matrix for LLMs is computationally prohibitive in practice [R1, R2]. To address this, we conducted new experiments to estimate the quantities listed in Table 1 using random subspaces, which offers a computationally feasible proxy. Please refer to the Global Response and Appendix F.7 of the revision for detailed results.
>
> [R1] Martens, J. (2010, June). Deep learning via Hessian-free optimization. In Proceedings of the 27th International Conference on International Conference on Machine Learning (pp. 735-742).
>
> [R2] Ghorbani, B., Krishnan, S., & Xiao, Y. (2019, May). An investigation into neural net optimization via hessian eigenvalue density. In International Conference on Machine Learning (pp. 2232-2241). PMLR.

---

> ### Author Response · Authors · 2025-11-19
> **Response for W3, W4**
>
> ### **W3: About Comparison with FL+PEFT.**
>
> **Our response:** We appreciate the reviewer’s recognition that our primary focus is on federated full-parameter tuning, which is the setting addressed by the main experimental results. At the same time, in Appendix E.5, we also evaluated the Parameter-Efficient Fine-Tuning (PEFT) regime and newly added HiSo+LoRA results in Table 8 of our revision, where we reported final accuracy and communication cost for several first-order and zeroth-order PEFT baselines.
>
> We agree with the reviewer that PEFT is an important direction for efficient LLM fine-tuning, and that a clearer comparison can help situate HiSo within the broader landscape. Following this suggestion, we have add the evaluation of memory usage.
>
> From Table 2 below, under the **full-parameter tuning** setting, we find that the memory cost of our HiSo is lower than first-order baselines (e.g., FedAvg and FedAdam), but it is higher than zeroth-order DeComFL due to the extra diagonal Hessian storage.
>
> Table 2: Client-Side Memeory Usage (OPT-125M model) for **full-parameter tuning**. The memory usage of a OPT-125M model is 478.96 MB (dtype=float32), which is the theoretical minimum baseline. We use this to calculate a factor for easier comparison.
> |Method|Single Client Memory Usage| N x ModelSize |
> |-|-|-|
> |FedAvg| 1530.21 MB| 3.2 |
> |FedAdam| 2487.40 MB| 5.2 |
> |DeComFL| 642.59 MB| 1.34 |
> |HiSo (ours)| 1263.93 MB| 2.68 |
>
> From Table 3 below, under the **PEFT** (i.e., LoRA) setting, we observe that the memory cost of our HiSo is lower than first-order baselines (e.g., FedAvg and FedAdam), but it is slightly higher than DeComFL.
>
> Table 3: Client-Side Memeory Usage (OPT-125M model) for **PEFT**. The memory usage of a OPT-125M model is 478.96 MB (dtype=float32). We use this to calculate a factor for easier comparison. If we employ LoRA (r=8), the memory usage of trainable parameters will be about 1 MB.
> |Method|Single Client Memory Usage| N x ModelSize | Note |
> |-|-|-|-|
> |FedAvg+LoRA (r=8)| 689.03 MB| 1.44 | Forward pass dominates.|
> |FedAdam+LoRA (r=8)| 689.03 MB| 1.44 | Forward pass dominates.|
> |DeComFL+LoRA (r=8)| 628.09 MB| 1.31 | DecomFL does not need to store activation, thus forward pass is not a dominant factor.|
> |HiSo+LoRA (ours) (r=8)| 629.21 MB| 1.32 | Slightly higer than DeComFL with LoRA because it needs to track hessian for LoRA parameters.|
>
> ---
>
> ### **W4: About hyperparameter sensitivity.**
>
> **Our response:** Thank you for your comment. We agree with the reviewer that the learning rate is a critical hyperparameter for all optimization algorithms. To ensure a fair and rigorous comparison, we adopted the standard practice of performing  the best learning rate for both of HiSo and other baseline algorithms. Since all first-order and zeroth-order methods are inherently sensitive to the learning rate, we prioritized the ablation study on the Hessian smoothing parameter $\nu$.
>
> We believe that the "exponential moving average decay factor" mentioned by the reviewer corresponds to the Hessian smoothing parameter $\nu$ in Eq.(12). In Figure 5, we has demonstrated that HiSo is robust to $\nu$ when it is close to 1 (the recommended setting). In the revision, we expanded this analysis to include the regime where $\nu \ll 1$ (see Appendix F.7.2). In this range, the estimator’s effective memory is reduced to only the most recent 1–2 rounds, yielding a high-variance Hessian approximation. As expected, performance deteriorates as $\nu$ decreases. This degradation confirms that our acceleration is indeed driven by the quality of the Hessian approximation. Practically, we recommend selecting $\nu \in [0.9, 1)$. Within this range, the algorithm remains robust and insensitive to the exact choice.

---

> ### Author Response · Authors · 2025-11-19
> **Response for Q2, Q3**
>
> ### **Q2: About HiSo + PEFT (e.g., LoRA).**
>
> **Our response:** Thank you for your question. Extending the Hessian-informed framework to Parameter-Efficient Fine-Tuning (PEFT) is a natural progression. While we omitted this discussion due to space constraints, we are pleased to share our preliminary thoughts on combining HiSo with PEFT below.
>
> 1. Direct Combination: From an optimization perspective, HiSo can be directly combined with methods like LoRA by restricting both the parameters and the learned Hessian to the low-rank additive modules. Our theoretical convergence bounds would apply directly to this restricted parameter space. Since the number of tunable parameters in PEFT is significantly smaller, the memory overhead for the Hessian is proportionally reduced, and we expect the Hessian approximation to be easier and faster to learn. However, this approach inherits the inherent performance ceiling of PEFT, and the relative communication advantage of HiSo (compared to first-order methods) may be less pronounced since PEFT already reduces transmission payloads.
>
> 2. Exploiting Low-Rank Structure: A more advanced approach would explicitly leverage the structural properties of PEFT within the zeroth-order framework. Taking LoRA ($W + U^TV$) as an example, rather than flattening the parameters of $U$ and $V$ and treating them uniformly, we should consider the matrix multiplication structure. Recent work suggests an asymmetric sensitivity between $U$ and $V$, where the matrix mapping inputs to the latent space ($V$) often plays a more critical role [R1]. A structure-aware HiSo could therefore apply a finer-grained Hessian approximation to $V$ while updating $U$ less aggressively. We consider this a promising direction for future investigation.
>
> [R1] Chen, Yiming, et al. "Enhancing zeroth-order fine-tuning for language models with low-rank structures." arXiv preprint arXiv:2410.07698 (2024).
>
> ---
>
> ### **Q3: Regarding the phenomenon of performance degrading to DeComFL.**
>
> **Our response:** Thank you for the thoughtful comment. In practice, we did not observe noticeable performance degradation when using proper hyper-parameters, even in the early stages of training. This is largely because the diagonal Hessian approximation is updated via exponential moving averaging (EMA), which stabilizes quickly and provides a sufficiently informative curvature signal after only a few rounds. Across all tasks, our HiSo consistently exhibited faster convergence. However, we did observe that using an excessively small EMA smoothing parameter (e.g., $\nu=0.1$), which results in a high-variance and noisy Hessian approximation, can lead to degraded performance comparable to DeComFL. In response, we have added a discussion of this phenomenon in Appendix F.7.2 of the revision.

---

> ### Comment · Reviewer_3tFi · 2025-11-27
> **Guarantee of Diagonal Preconditioner**
>
> Thank you for the rebuttal. I appreciate the authors’ effort in addressing my concerns including computational cost, validation of theoretical assumption, comparison with FL + PEFT, and hyper-parameter sensitivity.
>
> However, a key issue remains unresolved: **how to guarantee that the diagonal preconditioner actually improves convergence or generalization across different problem settings**. The rebuttal mentions that hyperparameters (e.g., damping coefficient, learning rate scaling) are tuned on a validation set, but this does not constitute a *principled guarantee*—especially in low-data or non-stationary regimes where validation tuning may be unreliable.
>
> Specifically, I still have the following question:
>
> > **What theoretical or empirical criteria guide the selection of hyperparameters (e.g., damping strength, step-size schedule) to ensure the preconditioner remains well-conditioned and beneficial?**
> > Without such guidance, the method risks being fragile in practice—effectively shifting the difficulty from optimization to hyperparameter tuning.
>
> If the authors can provide either (a) a stability analysis linking hyperparameter choices to curvature properties, or (b) a robust auto-tuning strategy (even heuristically motivated), it would significantly strengthen the paper’s practical contribution. As it stands, the current response alleviates some concerns but does not fully address the core issue of *reliable and predictable preconditioning*.

---

> > ### Author Response · Authors · 2025-11-28
> >
> > We sincerely thank the reviewer for pushing for a more principled approach to hyper-parameter selection. We acknowledge that it is a common concern that new algorithms often shift difficulty from optimization to tuning.
> > In fact, our paper provided both theoretical and empirical criteria guide for the hyper-parameter choices. We will reiterate it as follows:
> >
> > 1. Theoretical Criteria for Step-Size (Scale Invariance): The core benefit of Hessian-informed preconditioning is scale invariance. In standard ZO gradient descent, the optimal learning rate $\eta$ is inversely proportional to the Lipschitz constant $L$ (curvature) of the specific landscape ($\eta \propto 1/L$). Since $L$ varies wildly across tasks, tuning is fragile. In contrast, HiSo approximates the inverse Hessian $H^{-1}$. Since $H \approx L$, the preconditioned update scales as $H^{-1} \nabla f \approx (1/L) \cdot \nabla f$. This implicitly cancels out the scale of the problem. Consequently, the optimal learning rate for HiSo falls within a narrow, predictable range (typically $10^{-3}$ to $10^{-4}$) across diverse settings, eliminating the need for extensive task-specific tuning required by ZO-SGD.
> >
> > 2. Stability Criterion for the Smoothing Parameter $\nu$ (Stability Analysis): Regarding the smoothing parameter $\nu$, our theoretical analysis and new experiments (Appendix F.7) provide a clear stability criterion: $\nu$ must be large enough to ensure the effective memory length covers the local curvature variability. Empirically, we identified a "robust region" of $\nu \in [0.9, 0.999]$. As long as $\nu$ falls within this range, the estimator is well-conditioned. This serves as a robust heuristic: practitioners do not need to fine-tune $\nu$; they can simply set a default of $\nu=0.99$.
> >
> > 3. Practical Robustness Parallels Adam’s Default Hyper-parameters: This situation parallels the practical usage of Adam. While Adam introduces hyper-parameters $\beta_1$ and $\beta_2$, practitioners rarely tune them, relying instead on standard defaults $\beta_1=0.9$ and $\beta_2=0.99$ because performance is relatively insensitive to exact values. HiSo follows this paradigm: the second-order approximation is robust enough that default parameters achieve strong performance without intricate tuning.
> >
> > 4. Advanced Diagnostics: Finally, regarding a specific heuristic for guaranteeing success: for scenarios where computational budget allows, the random subspace metric we introduced in the Global Response can serve as a diagnostic tool. By verifying the spectral properties in a subspace, practitioners can confirm the approximation quality, though we believe the default settings are sufficient for most practical cases.
> >
> > In short, our paper provided both theoretical and empirical criteria guide for the  hyperparameter choices by (i) scale invariance for step-size, (ii) a well-defined stability region for the smoothing parameter, and (iii) empirical robustness comparable to well-established optimizers like Adam. We hope this clarifies that HiSo does not introduce additional hyper-parameter issue; rather, it yields robust and reliable optimization behavior across diverse problem settings.

---

### Official Review · Reviewer_FQxG · 2025-11-03

**Soundness:** 4
**Presentation:** 3
**Contribution:** 4
**Rating:** 6
**Confidence:** 3

**Summary:**

In this paper, the authors propose HiSo, a Hessian informed, scalar only, federated zeroth order optimizer for large language model fine tuning.  The authors first generalize the scalar only paradigm into a reusable federated optimization framework, where the server and clients exchange only scalar update codes and can reconstruct each other's parameter trajectories without ever sending high dimensional tensors. Within that framework, they introduce HiSo, which maintains a global diagonal Hessian approximation on the server and uses it to precondition the zeroth order perturbation directions. Effectively, HiSo samples Hessian aware update directions, similar in spirit to natural gradient or approximate Newton steps, but still communicates only scalars.

**Strengths:**

1. Theoretical contribution:  The paper proves non convex convergence bounds where the rate depends on a whitening rank related to the effective Hessian spectrum instead of the raw model dimension, and extends DeComFL style theory to multiple local steps per round.

2. Strong and concrete motivation:  Existing scalar only ZO methods solve bandwidth but converge painfully slowly. HiSo squarely targets this convergence bottleneck without giving up the scalar only advantage.

3. The paper repeatedly reports actual bandwidth numbers in KB and TB, and highlights 10^7 to 10^8 fold savings compared to FedAdam and FedAvg.

**Weaknesses:**

1. Benchmark scale and diversity：The main LLM experiments involve six clients with two sampled per round, and tasks are classification and extractive QA. Although these are standard NLP benchmarks and good stress tests for convergence and accuracy, they are still small compared to industrial federated networks across hospitals, phones, or enterprises. The paper would be stronger if it included either larger federations or at least a stress test with many more clients and skew patterns.


2. Backbone model: The authors only conduct experiments on OPT series models, how about the Qwen series models? I am curious about the performance of proposed method on those models.

3. Theory assumptions and coverage:
   The convergence guarantee depends on a low effective rank Hessian spectrum and on the quality of the diagonal Hessian approximation. While the paper provides evidence that the estimated diagonal Hessian has a long tail and argues that large language models empirically satisfy this, it does not show failure cases or quantify how often the whitening rank assumption holds in domains beyond language. This makes it harder to judge how robust the dimension independent claim really is.

**Questions:**

see weakness 1,2,3

---

> ### Author Response · Authors · 2025-11-19
> **Response for W1, W2**
>
> ### **W1: About benchmark scale and diversity.**
>
> **Our response:** Thank you for your constructive comment. We agree that the experiments with more clients and benchmarks will enhance the evaluation of our HiSo. Following the reviewer's suggestion, we have added several extra experiment results in the following Tables 1 and 2.
>
> (1) **More clients**. We execute extra experiments on OPT125M with **64 clients**, where only 25% clients participate in each round. Due to limited GPU resourses, we were unable to simultaneously scale to both larger client populations and larger LLMs. (2) **More datasets**. In addition to SST2, QQP and SQuAD, we added new datasets, such as WSC and WIC, covering broader NLP tasks. (3) **More patterns**. We added more experiments with different degrees of non-iid data controlled by Dirichlet parameter $\alpha$ to evaluate robustness under stronger data skew.
>
> Across these new settings, Tables 1 and 2 show that HiSo consistently outperforms DeComFL in test accuracy.
>
> Moreover, we actually added many other experiments about newer LLMs (e.g., Qwen and Gemma) to further demonstrate the generality of our HiSo.
>
> Table 1: Test Accuracy (64 clients, 25% participation ratio, $\alpha=1$)
> |Model|Dataset|DeComFL|HiSo|
> |-|-|-|-|
> |OPT-125M|SST2|84.75% $\pm$ 0.25|85.58% $\pm$ 0.29|
> |OPT-125M|QQP|60.13% $\pm$ 0.11|60.85% $\pm$ 0.14|
> |OPT-125M|SQuAD|34.21% $\pm$ 0.13|35.28% $\pm$ 0.10|
> |OPT-125M|WSC|58.91% $\pm$ 0.13|59.22% $\pm$ 0.09|
> |OPT-125M|WIC|52.87% $\pm$ 0.10|53.05% $\pm$ 0.13|
>
> Table 2: Test Accuracy (64 clients, 25% participation ratio, $\alpha=0.1$)
> |Model|Dataset|DeComFL|HiSo|
> |-|-|-|-|
> |OPT-125M|SST2|82.05% $\pm$ 0.21|82.77% $\pm$ 0.27|
> |OPT-125M|QQP|56.38% $\pm$ 0.23|57.45% $\pm$ 0.21|
> |OPT-125M|SQuAD|31.34% $\pm$ 0.23|32.73% $\pm$ 0.17|
> |OPT-125M|WSC|54.27% $\pm$ 0.14|54.95% $\pm$ 0.20|
> |OPT-125M|WIC|49.10% $\pm$ 0.15|50.06% $\pm$ 0.21|
>
> ---
>
> ### **W2: Experiment results on Qwen model.**
>
> **Our response:** Thank you for your question. We executed a series of experiments on the Qwen model and show the results as follows. Moreover, besides Qwen, we also executed experiments on the Gemma model (see Table 5 in Appendix E.3).
>
> From the following Table 3, we observe that our HiSo consistently achieves $1.6 \sim 2.5\times$ convergence speedup while reducing communication cost due to less required rounds.
>
> From the following Table 4, we find that our HiSo achieves higher test accuracy than all ZO baselines and achieves lowest communication cost among all baselines.
>
> Table 3: HiSo's Acceleration Performance on **Qwen3-0.6B**.
> Dataset|Method|Round|Speedup|Comm. Cost|
> |-|-|-|-|-|
> |SST-2|DeComFL|2500|$1\times$|83.16 KB|
> |SST-2|HiSo|1000|$2.5\times$|33.26 KB|
> |QQP|DeComFL|1050|$1\times$|34.93 KB|
> |QQP|HiSo|650|$1.6\times$|21.62 KB|
> |SQuAD|DeComFL|375|$1\times$|12.47 KB|
> |SQuAD|HiSo|225|$1.7\times$|7.48 KB|
>
> Table 4: Test Accuracy and Communication Cost Comparison on **Qwen3-0.6B**.
> |Method|SST-2|
> |-|-|
> |FedAvg|90.03\% $\pm$ 0.27 (1.68 TB)|||
> |FedAdam|90.12\% $\pm$ 0.21 (1.01 TB)|||
> |FedYogi|90.05\% $\pm$ 0.23 (1.08 TB)|||
> |FedAdagrad|89.89\% $\pm$ 0.21 (1.13 TB)|||
> |FedZO|84.43\% $\pm$ 0.21 (12.00 TB)|||
> |DeComFL|84.79\% $\pm$ 0.17 (83.16 KB)|||
> |HiSo(ours)|85.25\% $\pm$ 0.19 (33.56 KB)|||

---

> ### Author Response · Authors · 2025-11-19
> **Response for W3**
>
> ### **W3: Theory assumptions and coverage.**
>
> **Our response:** We thank the reviewer for raising these important points regarding robustness and failure modes.
>
> For the first part about low effective rank & approximation quality, please refer to our Global Response, where we address the theoretical validity of the low-effective rank assumption and the diagonal approximation in detail.
>
> For the second part about the failure mode: to address the concern regarding failure cases, we designed a new experiment testing the sensitivity of the diagonal Hessian approximation to the smoothing parameter $\nu$ (see Appendix F.7.2).
>
> We notice the parameter $\nu$ controls the effective history length used to estimate the Hessian. When $\nu < 0.5$, the estimator effectively utilizes only the most recent 1–2 rounds of zeroth-order information, resulting in a high-variance, uninformative Hessian approximation. In our original submission (Figure 5), we used $\nu \in [0.9, 0.95, 0.99]$, which ensures sufficient smoothing and stable performance. In the new experiment, we tested $\nu \in [0.1, 0.3, 0.4, 0.5, 0.8]$. As expected, performance deteriorates as $\nu$ decreases, confirming that a well-accumulated Hessian estimate is crucial for acceleration. Furthermore, while performance degrades with a poor Hessian approximation, the algorithm still converge to some reasonable performance. As discussed in Appendix F.7 "WHEN WELL-APPROXIMATION OF HESSIAN MATRIX DOES NOT HOLD", since the preconditioner $H$ remains a positive definite diagonal matrix, the "worst-case" scenario resembles a poorly scaled first-order method rather than a catastrophic failure. Conversely, this experiment validates that when the Hessian is well-learned (high $\nu$), it provides meaningful acceleration.

---

> > ### Comment · Reviewer_FQxG · 2025-11-27
> >
> > The additional experiments provided by the authors have addressed my concern. I will keep my positive score.

---

> > > ### Author Response · Authors · 2025-11-27
> > >
> > > Thank you so much for your positive feedback. We are pleased that the additional experiments successfully addressed your concern. If your have other questions, we are happy to provide further explanation.

---

### Official Review · Reviewer_HtDg · 2025-11-06

**Soundness:** 3
**Presentation:** 3
**Contribution:** 2
**Rating:** 4
**Confidence:** 3

**Summary:**

This paper studies federated learning via zeroth-order optimization. By proposing a method that preserves scalar-only communication and avoids transmitting second-order information, it significantly reduces computational costs. Theoretically, they demonstrates an accelerated convergence rate under a suitable Hessian structure.

**Strengths:**

The paper is well-written, with a well-motivated research goal and a clear description of the  algorithm.

**Weaknesses:**

I have the following concerns regarding the paper:

Theoretical Practicality and Depth: The theoretical analysis relies on a good approximation of the Hessian, yet the method employed in practice is only a diagonal approximation. This gap makes it difficult to appreciate the practical relevance of Theorem 1. Furthermore, a simple non-convex analysis seems insufficient, as it fails to capture the specific landscape properties of neural network loss functions.

Experimental Comprehensiveness: The experimental validation appears somewhat limited. It would be strengthened by including plots of the training loss against both iterations and wall-clock time on more experiments. Additionally, experiments on more datasets and with larger models would better demonstrate the scalability and robustness of the proposed method.

Novelty of Insight: The core idea seems limited. Since zeroth-order optimization inherently accesses only scalar information at each step, the advantage of communicating solely scalars appears straightforward. I did not find significant novel algorithmic insights in the current work.

**Questions:**

See the weakness.

---

> ### Author Response · Authors · 2025-11-19
> **Response for W1, W2**
>
> ### **W1: About theoretical practicality and depth.**
>
> **Our response:** We thank the reviewer for these insightful comments. We agree that bridging the gap between theoretical assumptions (well-approximated Hessian) and practical implementation (diagonal approximation) is central to understanding the algorithm's success.
>
> 1. On the Diagonal Approximation: While a diagonal matrix cannot capture the full geometry of a non-diagonal Hessian, it is often sufficient for spectral whitening, which is the key driver of convergence in our analysis. In high-dimensional NN landscapes, the Hessian is often diagonally dominant or has a bulk of eigenvalues that can be effectively clustered by diagonal scaling. To empirically validate this connection, we have added a new experiment (see Appendix F.7.3) analyzing the exact Hessian in restricted subspaces. We observe that our learned diagonal Hessian effectively reduces the condition number (whitening) within these subspaces, aligning well with the "well-approximation" condition in our theorem. This provides concrete evidence that the practical diagonal approximation satisfies the theoretical requirements for acceleration. See global response for more discussion as well.
>
> 2. On Non-Convex Analysis and NN Landscapes: We respectfully clarify that providing a convergence proof for general non-convex smooth functions is the standard rigor for optimization algorithms (e.g., Adam/AdamW, LoRa, FedAvg, Shampoo, etc.). Analyzing specific NN landscape properties (e.g., proving specific global minima existence) is an open problem in Learning Theory and is generally beyond the scope of Optimization algorithm design.
>
> 3. Furthermore, our analysis goes beyond "simple" non-convex bounds. The key insight is we combined the crucial forth-moment of diagonal weighted Gaussian (Lemma 1) into the optimization analysis framework. We further explicitly introduce the Low Effective Rank assumption to capture a specific property of NN landscapes—that the Hessian spectrum is heavy-tailed with many near-zero eigenvalues. By incorporating this landscape-specific property into our bounds (via $\kappa$ and $\zeta$), we provide a theoretical plausible explanation for why HiSo performs well, bridging the gap the reviewer highlighted. For this reason, we are unsure what additional "specific landscape properties" the reviewer believes should be incorporated. We would appreciate further clarification if above explanation is not sufficient.
>
> ---
>
> ### **W2: About experimental comprehensiveness.**
>
> **Our response:** Thank you for your comment.
>
> (1) We added new figures (**training loss vs iterations**, and **training loss vs wall-clock time**) in Figure 8 in Appendix E.6 in our revision. Those figures were plotted based on our experiment results Qwen3-0.6B model and SST-2 dataset. From Figure 8, HiSo consistently demonstrates substantially faster progress than DeComFL in both iterations and wall-clock time. On SST-2 with Qwen-3-0.6B, DeComFL requires roughly 2500 iterations to converge, while HiSo reaches the same accuracy within only about 1000 iterations, achieving a 2.5$\times$ speedup in optimization rounds. This faster iteration-level convergence directly translates into end-to-end efficiency: when measuring real execution time, including communication and computation, HiSo is about 2.1$\times$ faster than DeComFL in reaching the same final accuracy. The accuracy–time and loss–time curves further confirm that HiSo maintains a steeper descent trajectory throughout training, reducing both the number of updates and the total runtime needed for convergence. Overall, these results show that HiSo accelerates federated fine-tuning in both algorithmic efficiency (iterations) and practical efficiency (wall-clock time), providing significantly faster convergence under the same hardware and network conditions.
>
> (2) To further strengthen our experimental evaluation, we additionally include new LLMs such as Qwen and Gemma, as well as new datasets including WIC and WSC. Currently, our experiments span a broad range of model sizes from 125M, 270M, 350M, 0.6B, to 1.3B parameters, which provides a substantial scale variation. This range sufficiently demonstrates the scalability and consistent performance of HiSo across models of increasing capacity.

---

> ### Author Response · Authors · 2025-11-19
> **Response for W3**
>
> ### **W3: About novelty of insight.**
>
> **Our response:** Thank you for your comment. We believe that there is a misunderstanding of ZO optimization and our HiSo. We would like to explain several challenges of our ZO algorithm design.
>
> (1) Scalar-only communication is nontrivial, especially for the downlink direction. Specifically, the ZO gradient can be decomposed into two parts: **a gradient scalar $g$** and a perturbation vector generated by **a random seed $s$** (also a scalar). This decomposition can make uplink communication (from clients to the server) achieve scalar-only communication because the participating clients only need to send their local gradient scalars back to the server. **However, the tricky part is downlink communication (from the server to sampled clients)**. Since it is almost impossible for a client to participate in every round continuously in practice, the server needs to store global gradient scalar+seed pairs, and each client's participation history needs to be recorded for clients' local model reconstruction. By the way, the memeory cost for storing these data is quite cheap, which we discussed in Appendix E.2. Besides, after the local update steps, the client needs to revert their updated local model back to the initial model of the current round (shown in Figure 2 in our paper).
>
> (2) Utilizing ZO information to estimate Hessian is itself challenging. Please notice that we consider the FL scenario that contains dozens of clients that do not share the local information. In FL, it is common to have partial client participation and local update. Both are not the standard operation that can trivially extend it to obtained the Hessian meaningfully.
>
> (3) Making a Hessian-informed ZO method compatible with scalar-only communication is nontrivial as well. Incorporating curvature typically requires transmitting Hessian-related information, which conflicts with the goal of scalar-only communication. Our work resolves this tension by designing a mechanism in which the curvature enters locally through a diagonal estimator, while the communication pattern remains strictly scalar-only. Achieving convergence acceleration without increasing communication dimensionality is a key novelty that, to our knowledge, has not been explored in prior ZO-FL literature.

---

### Official Review · Reviewer_NVun · 2025-11-06

**Soundness:** 4
**Presentation:** 4
**Contribution:** 3
**Rating:** 6
**Confidence:** 5

**Summary:**

The authors of this manuscript propose HiSo, a Hessian-informed federated zeroth-order (ZO) optimization method. It aims to accelerate the convergence of ZO-FL, which is attractive for LLM fine-tuning due to its scalar-only (dimension-free) communication, but typically suffers from slow convergence. The method computes a global diagonal Hessian approximation using an Adam-style update rule, which does not require transmitting any second-order information. The authors use this approximation to inform the ZO update direction. They provide a theoretical analysis showing convergence rates independent of model dimension and empirical results demonstrating a 1-5x speedup in communication rounds over the DeComFL baseline.

**Strengths:**

1- The improved convergence by HiSo is significant in comparison to other ZO-FL benchmarks, while keeping the communication costs significantly small. The 1-5x speedup over DeComFL is a practical and valuable improvement.

2- The paper provides a theoretical analysis to support the method, proving a convergence rate independent of model dimension `d` for non-convex functions.

3- The proposed method addresses a clear and important bottleneck in federated LLM fine-tuning, namely the high communication cost of traditional first-order methods and the slow convergence of previous ZO methods.

**Weaknesses:**

1- The experimental validation is limited to the OPT family of models (OPT-125M to OPT-2.7B). These models are somewhat dated and are known to be undertrained. The effectiveness of HiSo on newer, more capable models (e.g., smaller variants of LLaMA-3.2, Qwen-2.5, Gemma 3, or SmolLM) is not demonstrated. It is unclear if the same optimization behavior will hold on these new architectures.

2- The claim that HiSo is a "Hessian-informed" or "second-order" method is potentially misleading. The proposed update for the diagonal Hessian approximation in Equation (12) is a recursive exponential moving average of the squared updates. This formulation is functionally identical to the variance/second-moment tracking in first-order adaptive optimizers like RMSProp or Adam. This makes the contribution appear more as an application of an adaptive preconditioner to the ZO setting, rather than a true second-order method.

**Questions:**

1- How does the performance of HiSo change when applied to newer generations of models, such as the HuggingFace SmolLM or smaller variants of Qwen-2.5, LLaMA-3.2, and Gemma 3? Do the same convergence benefits hold?

2- The authors call the method "Hessian-informed." However, the update in Equation (12) is mathematically very similar to the variance/second-moment update in the Adam optimizer (a first-order method). Could the authors clarify why this should be considered a second-order method and not a first-order adaptive method? A direct comparison of the HiSo formulation with Adam would be insightful.

3- The paper focuses on fine-tuning. Can the proposed ZO-FL framework be used for the pretraining of LLMs? What are the primary challenges in applying this method to the pretraining regime? I understand pretraining is expensive and do not expect experiments, but insight into the challenges would be helpful.

---

> ### Author Response · Authors · 2025-11-19
> **Response for W1&Q1**
>
> ### **W1&Q1: Experiments on newer LLMs.**
>
> **Our response:** Thank you for the comment. We agree with the reviewer that evaluating our HiSo on more recent architectures can further strengthens the robustness of our claims. Following the reviewer’s suggestion, we have executed experiments on newer models, including Qwen-3 and Gemma-3. We show the results in Table 1, 2, 3 below. Across all datasets (SST-2, QQP, and SQuAD), our HiSo consistently delivers convergence acceleration (1.4×–2.6×) and maintains communication cost in the kilobyte range, mirroring the improvements observed on the OPT family. These additional results indicate that HiSo generalizes well beyond the OPT models and remains effective across modern and diverse LLM architectures. In addition, we are currently running further experiments on SmolLM-3B models, and we will include the results once we get the results.
>
> Table 1: HiSo's Acceleration Performance on SST2 Dataset.
> |Model|Method|Round|Speedup|Comm. Cost|
> |-|-|-|-|-|
> |Qwen3-0.6B|DeComFL|2500|$1\times$|83.16 KB|
> |Qwen3-0.6B|HiSo|1000|$2.5\times$|33.26 KB|
> |Gemma-3-270M|DeComFL|2125|$1\times$|70.60 KB|
> |Gemma-3-270M|HiSo|1025|$2.1\times$|34.05 KB|
>
> Table 2: HiSo's Acceleration Performance on QQP Dataset.
> |Model|Method|Round|Speedup|Comm. Cost|
> |-|-|-|-|-|
> |Qwen3-0.6B|DeComFL|1050|$1\times$|34.93 KB|
> |Qwen3-0.6B|HiSo|650|$1.6\times$|21.62 KB|
> |Gemma-3-270M|DeComFL|850|$1\times$|28.27 KB|
> |Gemma-3-270M|HiSo|625|$1.4\times$|20.79 KB|
>
> Table 3: HiSo's Acceleration Performance on SQuAD Dataset.
> |Model|Method|Round|Speedup|Comm. Cost|
> |-|-|-|-|-|
> |Qwen3-0.6B|DeComFL|375|$1\times$|12.47 KB|
> |Qwen3-0.6B|HiSo|225|$1.7\times$|7.48 KB|
> |Gemma-3-270M|DeComFL|900|$1\times$|29.94 KB|
> |Gemma-3-270M|HiSo|450|$2\times$|14.97 KB|
>
> Table 4: Comparison of Test Accuracy and Communication Cost. We use five perturbations for all ZO methods.
> |Model|Method|SST-2|
> |-|-|-|
> |Qwen3-0.6B|FedAvg|90.03\% $\pm$ 0.27 (1.68 TB)|||
> |Qwen3-0.6B|FedAdam|90.12\% $\pm$ 0.21 (1.01 TB)|||
> |Qwen3-0.6B|FedYogi|90.05\% $\pm$ 0.23 (1.08 TB)|||
> |Qwen3-0.6B|FedAdagrad|89.89\% $\pm$ 0.21 (1.13 TB)|||
> |Qwen3-0.6B|FedZO|84.43\% $\pm$ 0.21 (12.00 TB)|||
> |Qwen3-0.6B|DeComFL|84.79\% $\pm$ 0.17 (83.16 KB)|||
> |Qwen3-0.6B|HiSo(ours)|85.25\% $\pm$ 0.19 (33.56 KB)|||
> |Gemma-3-270M|FedAvg|87.14\% $\pm$ 0.23 (0.76 TB)|||
> |Gemma-3-270M|FedAdam|87.43\% $\pm$ 0.19 (0.46 TB)|||
> |Gemma-3-270M|FedYogi|87.38\% $\pm$ 0.20 (0.50 TB)|||
> |Gemma-3-270M|FedAdagrad|87.05\% $\pm$ 0.25 (0.45 TB)|||
> |Gemma-3-270M|FedZO|83.93\% $\pm$ 0.21 (4.54 TB)|||
> |Gemma-3-270M|DeComFL|83.85\% $\pm$ 0.15 (70.60 KB)|||
> |Gemma-3-270M|HiSo (ours)|84.28\% $\pm$ 0.18 (34.88 KB)|||

---

> > ### Author Response · Authors · 2025-11-19
> > **Response for W2&Q2**
> >
> > ### **W2&Q2: About using "Hessian-informed" and "second-order" to describe our method in this paper.**
> >
> > **Our response:** We sincerely appreciate the reviewer raising this point regarding the terminology "Hessian-informed" versus "second-order." We agree that the distinction is subtle and are happy to clarify. Strictly speaking, HiSo is a distributed zeroth-order optimization algorithm, but its expected update direction approximates the Newton direction (the product of the inverse Hessian and the gradient).
> >
> > 1. "Second-Order" vs. "Hessian-Informed": A "Second-Order Optimization Algorithm" typically refers to methods that explicitly compute or query the Hessian matrix (or Hessian-vector products). HiSo relies solely on function evaluations and is therefore strictly a Zeroth-Order (ZO) algorithm, as stated in our title. We have reviewed the manuscript to ensure we do not claim it is a classical second-order algorithm.However, we use the term "Hessian-informed" because the method approximates curvature information to guide the update. This is analogous to Quasi-Newton methods (like L-BFGS): they are technically First-Order algorithms, yet they utilize "second-order information" by approximating the Hessian. Similarly, HiSo is a ZO algorithm that approximates the Hessian diagonal to precondition the update. Furthermore, this terminology is standard in the zeroth-order optimization literature [R1, R2].
> >
> > 2. Resemblance to Adam/RMSProp: We are fully aware of the structural resemblance between our update rule and adaptive gradient methods. We explicitly acknowledged this in our "Related Work" starting with Adaptive Gradient method,  in Footnote 1 (Page 5): "More accurately, our method resembles RMSProp...", and FedAdam/FedYogi are our key baselines in the experiments. Conceptually, just as Adam is a first-order algorithm that leverages curvature-like information (via second moments), HiSo is the corresponding zeroth-order counterpart. A key distinction, however, is that HiSo achieves this in a distributed setting without the dimension-dependent communication costs that are typically prohibitive for first-order methods.
> >
> > 3. Framework Flexibility: The adoption of the specific diagonal update (similar to RMSProp) is primarily driven by practical considerations (minimizing compute/memory). Notice our theoretical analysis just relies on the whitened Hessian trace, $\text{Tr}(H^{-1/2} \Sigma H^{-1/2})$, and is not intrinsically tied to the specific Adam/RMSProp formulation. The HiSo framework is general: if a better method exists to proxy the Hessian curvature while maintaining minimal resource requirements, HiSo can be easily adapted to utilize it.
> >
> > Lastly, we have added a footnote at the first occurrence of "Hessian-informed" to clarify that this term in our paper implies preconditioning the ZO update to approximate the Hessian-inverse product, rather than explicit Hessian computation.
> >
> > [R1] Zhao, Yanjun, et al. "Second-Order Fine-Tuning without Pain for LLMs: A Hessian Informed Zeroth-Order Optimizer." The Thirteenth International Conference on Learning Representations, 2025.
> >
> > [R2] Ye, Haishan, et al. "Hessian-aware zeroth-order optimization." IEEE transactions on pattern analysis and machine intelligence (2025).

---

> ### Author Response · Authors · 2025-11-19
> **Response for Q3**
>
> ### **Q3: Discussion on using ZO-FL framework for LLM pretraining.**
>
> **Our response:** We thank the reviewer for this insightful question. A lot of existing works show that ZO optimization can be effectively applied to LLM fine-tuning. However, to the best of our knowledge, no existing ZO optimization framework has yet been successfully applied to the pretraining of LLMs. This gap arises because pretraining poses some challenges for ZO methods:
>
> **High-rank update of LLM pretraining vs low-rank update of fine-tuning**: In LLM pretraining, neural networks exhibit high-rank update dynamics over long trajectories [R1], or start with an initially high-rank update structure [R2]. By contrast, as a post-training stage, LLM fine-tuning exhabits low-rank update [R2, R3]. ZO methods usually need a large number of random perturbations to stably estimate gradients, and the number of perturbations required grows with the intrinsic rank of the update. Thus, in the high-rank landscape of pretraining, ZO would require an impractically large number of perturbations to obtain stable estimates. However, in the low-rank subspace induced by fine-tuning, even a small number of perturbations can approximate the relevant descent directions, making ZO feasible and efficient.
>
> Although applying ZO to LLM pretraining has not achieved currently, we regard extending ZO methods to pretraining as an promising future research direction.
>
> [R1] Lialin, V., Muckatira, S., Shivagunde, N., & Rumshisky, A. ReLoRA: High-Rank Training Through Low-Rank Updates. In The Twelfth International Conference on Learning Representations.
>
> [R2] Aghajanyan, A., Gupta, S., & Zettlemoyer, L. (2021, August). Intrinsic dimensionality explains the effectiveness of language model fine-tuning. In Proceedings of the 59th annual meeting of the association for computational linguistics and the 11th international joint conference on natural language processing (volume 1: long papers) (pp. 7319-7328).
>
> [R3] Li, C., Farkhoor, H., Liu, R., & Yosinski, J. (2018, February). Measuring the Intrinsic Dimension of Objective Landscapes. In International Conference on Learning Representations.

---

> > ### Comment · Reviewer_NVun · 2025-11-26
> >
> > Thank you for your clarifications. I would like to maintain my positive evaluation of your work.
> >
> > My main concern regarding the added results is that for more recent models (such Gemma3 and Qwen3) the accuracy gap between HiSo and other benchmarks can be much more significant than the older generation of models (e.g. OPT models initially reported in the paper).

---

> > > ### Author Response · Authors · 2025-11-27
> > > **Thank you for your reply.**
> > >
> > > Thank you very much for your thoughtful follow-up and for the positive evaluation of our work. Regarding the concern that the accuracy gap between HiSo and first-order baselines appears larger on newer model families (e.g., Gemma-3 and Qwen-3) than on the older OPT series, we would like to offer the following clarification.
> > >
> > > Our original experiments adopted the common perturbation setting ($P=5$), which is widely used in the ZO literature and has the stable convergence behavior, relatively low communication overhead, and decent accuracy in our experiments. This choice helped us establish a strict baseline focused on showing HiSo's core advantage: achieving accelerated convergence with quite low communication cost.
> > >
> > > It is well-established that increasing the number of perturbations improves the accuracy of ZO methods [R1, R2], and this trend is also supported by our theory (Theorem 2 in Appendix G). When we increase $P$, the accuracy gap between our HiSo and first-order baselines decreases consistently across all model families. Importantly, with larger $P$ (e.g., $P=20$), we observe no statistically meaningful performance differences between OPT, Gemma-3, and Qwen-3 models. This pattern is shown in Table 5, and a comprehensive comparison for all baselines is provided in Table 6 below.
> > >
> > > [R1] Liu, S., Chen, P. Y., Kailkhura, B., Zhang, G., Hero III, A. O., & Varshney, P. K. (2020). A primer on zeroth-order optimization in signal processing and machine learning: Principals, recent advances, and applications. IEEE Signal Processing Magazine, 37(5), 43-54.
> > >
> > > [R2] Li, Z., Ying, B., Liu, Z., Dong, C., & Yang, H. Achieving Dimension-Free Communication in Federated Learning via Zeroth-Order Optimization. In The Thirteenth International Conference on Learning Representations.
> > >
> > > Table 5: Test Accuracy Gap between HiSo and FO baselines. For the results of HiSo, ($\cdot$) denotes the accuracy gap calculated by (HiSo's acc - Average acc of all FO baselines above)
> > > |Method|Qwen3-0.6B + SST-2|Gemma-3-270M + SST-2|OPT-350M + SST-2|
> > > |-|-|-|-|
> > > |FedAvg|90.03\%|87.14\%|89.79\%|
> > > |FedAdam|90.12\%|87.43\%|89.92\%|
> > > |FedYogi|90.05\%|87.38\%|89.68\% |
> > > |FedAdagrad|89.89\%|87.05\%|87.42\%|
> > > |Averaged acc of all FO baselines above|90.02\%|87.25\%|89.20\%|
> > > |HiSo ($P=5$)|85.25\% (-4.77 \%)|84.28\% (-2.97\%)|87.50\% (-1.70\%)|
> > > |HiSo ($P=10$)|87.75\% (-2.27\%)|86.56\% (-0.69\%)|88.34\% (-0.86\%)|
> > > |HiSo ($P=20$)|88.58\% (-1.44\%)|86.89\% (-0.36)|89.12\% (-0.08\%)|
> > >
> > >
> > > Table 6: Comprehensive Comparison of Test Accuracy and Communication Cost. HiSo achieves the lowest communication cost.
> > > |Method|Qwen3-0.6B + SST-2|Gemma-3-270M + SST-2|OPT-350M + SST-2|
> > > |-|-|-|-|
> > > |FedAvg|90.03\% $\pm$ 0.27 (1.68 TB)|87.14\% $\pm$ 0.23 (0.76 TB)|89.79\% $\pm$ 0.05 (0.58 TB)|
> > > |FedAdam|90.12\% $\pm$ 0.21 (1.01 TB)|87.43\% $\pm$ 0.19 (0.46 TB)|89.92\% $\pm$ 0.20 (0.21 TB)
> > > |FedYogi|90.05\% $\pm$ 0.23 (1.08 TB)|87.38\% $\pm$ 0.20 (0.50 TB)|89.68\% $\pm$ 0.29 (0.25 TB)|
> > > |FedAdagrad|89.89\% $\pm$ 0.21 (1.13 TB)|87.05\% $\pm$ 0.25 (0.45 TB)|87.42\% $\pm$ 0.09 (0.23 TB)|
> > > |FedZO ($P=5$)|84.43\% $\pm$ 0.21 (12.00 TB)|83.93\% $\pm$ 0.21 (4.54 TB)|86.55\% $\pm$ 0.23 (0.68 TB)|
> > > |DeComFL ($P=5$)|84.79\% $\pm$ 0.17 (83.16 KB)|83.85\% $\pm$ 0.15 (70.60 KB)|86.72\% $\pm$ 0.28 (21.56 KB)|
> > > |HiSo ($P=5$)|85.25\% $\pm$ 0.19 (33.56 KB)|84.28\% $\pm$ 0.18 (34.88 KB)|87.50\% $\pm$ 0.22 (17.33 KB)|
> > > |FedZO ($P=10$)|85.35\% $\pm$ 0.20 (9.48 TB)|84.57\% $\pm$ 0.16 (3.72 TB)|86.93\% $\pm$ 0.20 (0.51 TB)|
> > > |DeComFL ($P=10$)|85.80\% $\pm$ 0.18 (49.22 KB)|84.60\% $\pm$ 0.15 (63.75 KB)|87.06\% $\pm$ 0.17 (69.33 KB)|
> > > |HiSo ($P=10$)|87.75\% $\pm$ 0.25 (35.16 KB)|86.56\% $\pm$ 0.20 (37.50 KB)|88.34 $\pm$ 0.16 (27.73 KB)|
> > > |FedZO ($P=20$)|86.77\% $\pm$ 0.15 (6.54 TB)|85.12\% $\pm$ 0.12 (3.17 TB)|87.75\% $\pm$ 0.14 (0.37 TB)|
> > > |DeComFL ($P=20$)|86.64\% $\pm$ 0.17 (111.56 KB)|86.29\% $\pm$ 0.13 (142.03 KB)|88.70\% $\pm$ 0.16 (62.40 KB)|
> > > |HiSo ($P=20$)|88.58\% $\pm$ 0.20 (61.98 KB)|86.89\% $\pm$ 0.16 (70.54. KB)|89.12\% $\pm$ 0.12 (41.60 KB)|

---

### Author Response · Authors · 2025-11-19
**Global Response**

Dear all reviewers,

We sincerely appreciate your constructive comments. We provide this global response to clarify the **diagonal Hessian learning strategy** and the **well-approximated Hessian assumption**, which appear in several reviews and are central to our contributions.

**(1) Our main theorem does not require well-approximated Hessian and low effective rank Hessian assumptions.**

Theorem 1 does not rely on low effective rank and well-approximated Hessian. Instead, theorem 1 explicitly shows the convergence behavior in the general case, where the quantities $\bar{\rho}$ and $\bar{\phi}$ capcture the impact of Hessian information, a key theoretical contribution of our work. When we introduce additional structure, such as the widely used low-effective-rank assumption or the well-approximated Hessian assumption proposed in this paper, $\bar{\rho}$ and $\bar{\phi}$ admit concise interpretations and tighter upper bounds. This yields the clean and faster convergence rates in Corollaries 1–3 and provides a principled explanation for the empirical acceleration observed in our experiments. In addition, we would like to point out that the original submission included a Remarks about well-approximated condition (Lines 402-407) and a dedicated discussion on when the well-approximated Hessian assumption may fail (Appendix F.7).

**(2) Our choice of a diagonal Hessian approximation is well-founded.**

First, extensive empirical evidence in the optimization and deep-learning literature shows that neural-network Hessians exhibit strong diagonal structure [R1]. In such settings, per-coordinate curvature magnitudes, which a diagonal matrix captures, are the dominant factors that influence optimization dynamics. Second, the diagonal structure is the curvature representation that is compatible with the scalar-only communication architecture.

**(3) New experiments for the gap between theoretical assumptions and Hessian measurements.**

We acknowledge the gap between assumptions on the Hessian and empirical verification at LLM scale. Computing full Hessians for LLMs is prohibitive in computation and memory ($O(d^3)$) [R2, R3]. To our knowledge, no prior work has directly evaluated full Hessians for LLM and existing empirical Hessian studies are limited to small models.

Nevertheless, to provide a more direct evidence, we designed a new approach that computes the quantities proposed in table 1. **Notice that, while exact computation for the full model is computational prohibitive, we can rigorously analyze the exact Hessian within restricted subspaces**. To do so, we randomly sampled 1000 parameters to compute the exact $1000 \times 1000$ Hessian matrix, $\hat{\Sigma}$. We then projected the learned Hessian $H$ from HiSo into this same subspace to obtain $\hat{H}$.

The eigenvalue correlation between $\hat{\Sigma}$ and $\hat{H}$ is just modest (5-20%, we think it is a reasonable value for diagonal approximations). But, when we evaluate the quantity $\mathbb{E}\|\hat{u}\|_{\hat{\Sigma}}$ discussed table 1, we observe much better improvements. The whitened Hessian eigenvalues are significantly smaller than the dimension $d$ and lower than the low effective rank estimation in that subspace. **Compared to the ideal synethatic result in figure 4, it is reasonable to expect the improvement is not that significiant. But this improvement is more or less consistent with our LLM experiment in Figure 7.** We repeated this process across multiple sampled subspaces; the boxplot in Figure 10 confirms the consistency of this observation.

Furthermore, we acknowledge that subspace Hessian eigenvalues are not unbiased estimators of the full Hessian. To mitigate this gap, we plotted the estimated quantities across varying subspace dimensions (Figure 10). As expected, the worst-case Lipschitz estimation $Ld$ increases rapidly with dimension, whereas the whitening metric remains robust and insensitive to the dimension change.

**Lastly, we added several more LLM experiments (e.g., Qwen and Gemma) based on various metrics in the revision**, and they provide more evidences that our diagonal Hessian approximation by zeroth-order gradients is successful.

Sincerely,

Authors


[R1] Dong, Zhaorui, et al. "Towards quantifying the hessian structure of neural networks." arXiv preprint arXiv:2505.02809 (2025).

[R2] Martens, J. (2010, June). Deep learning via Hessian-free optimization. In Proceedings of the 27th International Conference on International Conference on Machine Learning (pp. 735-742).

[R3] Ghorbani, B., Krishnan, S., & Xiao, Y. (2019, May). An investigation into neural net optimization via hessian eigenvalue density. In International Conference on Machine Learning (pp. 2232-2241). PMLR.

---

### Meta-Review · Area_Chair_Md6q · 2026-01-06

**Summary:**

This paper proposes HiSo, a novel Hessian-informed zeroth-order optimization method for federated learning that achieves dimension-free communication while accelerating convergence. The key innovation lies in leveraging global diagonal Hessian approximations as preconditioners without transmitting second-order information, thus preserving the scalar-only communication paradigm. Theoretically, the authors establish a convergence rate independent of model dimension and Lipschitz constant under certain Hessian approximation assumptions, providing the first such result for ZO methods in FL with multiple local updates. Empirically, HiSo demonstrates 1-5× speedup in communication rounds and up to 90 million times communication savings compared to first-order baselines across diverse LLM fine-tuning tasks.

**Reviewer Scores:**

No

---

### Decision · Program_Chairs · 2026-01-26

Accept (Poster)